# sigmoidF1: A Smooth F1 Score Surrogate Loss for Multilabel Classification

**Gabriel Bénédict**  *g.benedict@uva.nl*
*RTL Nederland B.V. & University of Amsterdam*

**Hendrik Vincent Koops**  *vincent.koops@rtl.nl*
*RTL Nederland B.V.*

**Daan Odijk**  *daan.odijk@rtl.nl*
*RTL Nederland B.V.*

**Maarten de Rijke**  *m.derijke@uva.nl*
*University of Amsterdam*

**Reviewed on OpenReview:** *https://openreview.net/forum?id=gvSHaaD2wQ*

## Abstract

Multilabel classification is the task of attributing multiple labels to examples via predictions. Current models formulate a reduction of the multilabel setting into either multiple binary classifications or multiclass classification, allowing for the use of existing loss functions (sigmoid, cross-entropy, logistic, etc.). These multilabel classification reductions do not accommodate for the prediction of varying numbers of labels per example. Moreover, the loss functions are distant estimates of the performance metrics. We propose *sigmoidF1*, a loss function that is an approximation of the macro F1 score that (i) is smooth and tractable for stochastic gradient descent at training time, (ii) naturally approximates a multilabel metric, and (iii) estimates both label suitability and label counts. We show that any confusion matrix metric can be formulated with a smooth surrogate. We evaluate the proposed loss function on text and image datasets, and with a variety of metrics, to account for the complexity of multilabel classification evaluation. sigmoidF1 outperforms other loss functions on one text and three image datasets over several metrics. These results show the effectiveness of using inference-time metrics as loss functions for non-trivial classification problems like multilabel classification.

## 1 Introduction

Many real-world classification problems are challenging because of unclear (or overlapping) class-boundaries, subjectivity issues, and disagreement between annotators. Multilabel learning tasks are common, e.g., document and text classification often deal with multilabel problems (Hull, 1994; Bruno et al., 2013; Yang, 2004; Blosseville et al., 1992), as do query classification (Kang & Kim, 2003; Manning et al., 2008), image classification (Shen et al., 2017; Xiao et al., 2010) and product classification (Amoualian et al., 2020). Existing optimization frameworks typically split the task into known problems and sum over existing losses $\sum \mathcal{L}_{\mathrm{MC}}$, with $\mathcal{L}_{\mathrm{MC}}$ any multiclass classification loss – oftentimes variations of the cross-entropy or logistic loss. Wydmuch et al. (2018) define these frameworks as *multilabel reduction* techniques; Menon et al. (2019) put an emphasis on two: One-Versus-All (OVA)[1] and Pick-All-Labels (PAL). For example, if $C$ is the number of possible classes, OVA and PAL reformulate the multilabel problem to $C$ binary classification and $C$ multiclass classification problems, respectively (see Section 2.3). These methods assume that, for one example,

---

[1]This was already described in (Dembczyński et al., 2010) and further formalized in (Wydmuch et al., 2018).

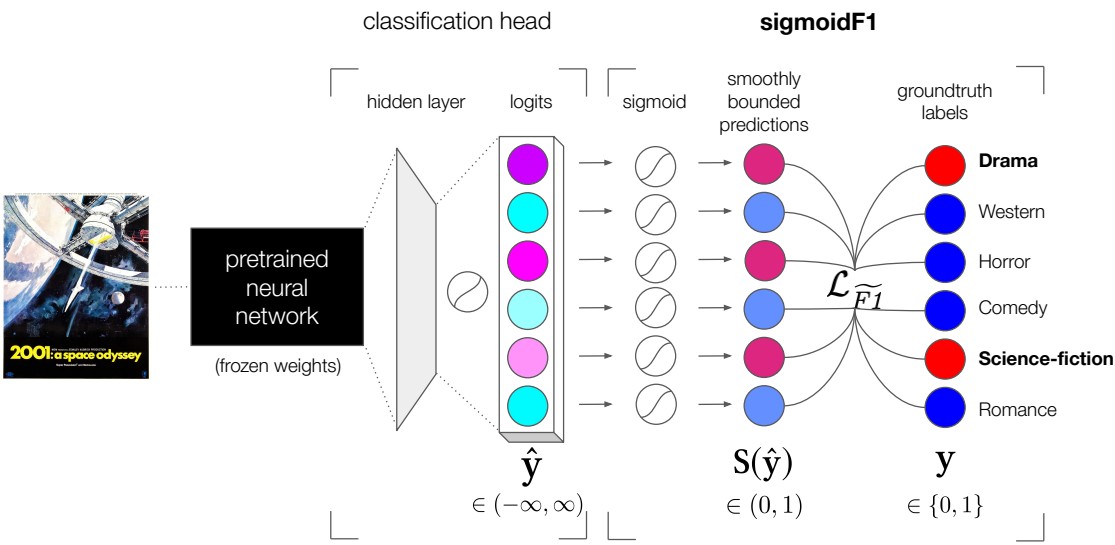

Figure 1: Experimental setup for *sigmoidF1* as a loss function for **multilabel classification**. Here, a movie poster image is fed to a pre-trained network with a custom classification head that outputs logits (i.e., unbounded values) for each class (i.e., movie genre). At training time, a sigmoid function forces logits towards either $-1$ or $1$, respectively negative and positive predictions (illustrated by the darker colors). Confusion matrix metrics and macro F1 can then subsequently be computed. Here, $S(\hat{y}_{horror})$ is close to 1, but the ground truth data claims that *2001: a space odyssey* is not a horror movie; this approximately corresponds to a false positive. Note that $\mathcal{L}_{\widetilde{F1}}$ is computed over a whole batch at training time as a macro measure with the formulas in Sections 4.3 and 4.4. With this setup, one can optimize directly for the metric of interest at training time. Our image and text classification tasks below show improved results when compared to existing losses.

label probabilities (a.k.a. Bayes Optimal Classifier (Dembczyński et al., 2010)) are marginally independent of other label probabilities. Menon et al. show mathematically and empirically that reduction methods (OVA and PAL) can optimize for precision or recall, but not for both precision and recall at once. More generally, a shortcoming shared by OVA and PAL is their reliance on the binary or multiclass classification setting and the lack of a pure multilabel approach – inspired by binary classification literature (see most recently (Gai et al., 2019) and their F1 surrogate loss functions on 3-layer neural networks). We are not aware of a metric surrogate loss function that deals with multilabel classification in a modern deep learning setting in a single task. Figure 1 illustrates our approach with a concrete example of classifying a movie poster into movie genres with a single loss function: *sigmoidF1*.

**Proposed solution to multilabel problems.** We propose a loss function $\mathcal{L}_{\widetilde{F1}}$ that (i) naturally approximates the macro $F1$ classification metric (see Table 3), (ii) estimates label probabilities and label counts (see Eq. 7), and (iii) is decomposable for stochastic gradient descent at training time (see Section 4.1 and Figure 2). Our proposed solution is to minimize a surrogate of the F1 metric as a loss. Strictly speaking, we minimize $1 - \widetilde{F1}$, where $\widetilde{F1}$ is a smooth version of $F1$. Using a metric as a loss function is unpopular for metrics that require a form of thresholding (e.g., counting the number of true positives), as minimizing a step loss function (a.k.a. 0-1 loss) is intractable. The soft margin for support vector machines is an early example, where the intractability of the direct 0-1 loss optimization is overcome with the hinge loss (Cortes & Vapnik, 1995). We resolve this by approximating the step function by a sigmoid curve (see Figure 1).

**Main contributions.** We introduce *sigmoidF1*, an F1 score surrogate, with a sigmoid function acting as a surrogate thresholding step function. *sigmoidF1* allows for the use of the F1 metric that simultaneously optimizes for label prediction and label counts in a single task. *sigmoidF1* is benchmarked against loss functions commonly used in multilabel learning and other existing multilabel models. We show that our

custom losses improve predictions over current solutions on several different metrics, across text and image classification tasks. PyTorch and TensorFlow source code are made available.[2]

## 2 Background

We use a traditional statistical framework as a guideline for multilabel classification methods (Tukey, 1977). We distinguish the desired theoretical statistic (the **estimand**), its functional form (the **estimator**) and its approximation (the **estimate**); estimates can be benchmarked with **metrics**. We show how multilabel reduction estimators tend to reformulate the estimand and treat labels as marginally independent. For example, by treating a multilabel problem as a succession of binary classification tasks. However, with a proper estimator, it is possible to directly model the estimand. If F1 score is indeed the statistic of interest (i.e. estimand), our proposed loss function, *sigmoidF1*, accommodates for the true estimand.

We define a learning algorithm $\mathcal{F}$ (i.e., a class of estimators) that maps inputs to outputs given a set of hyperparameters $\mathcal{F}(\cdot; \Theta) : \mathcal{X} \to \mathcal{Y}$. We consider a particular case, with the input vector $\mathbf{x} = \{x_1, \ldots, x_n\}$ and each observation is assigned $k$ labels (one or more) $\mathbf{l} = \{l_1, \ldots, l_C\}$ out of a set of $C$ classes. $y_i^j$ are binary variables, indicating presence of a label for each observation $i$ and class $j$. Together, they form the matrix output $\mathbf{Y}$. This is our multilabel setting. Note that multiclass classification can be considered as an instance of multilabel classification, where a single label is attributed to an example.

### 2.1 Estimand and definition of the risk

We distinguish between two scenarios: the *multiclass* and the *multilabel* scenario. In the multiclass scenario, a single example is attributed one class label (e.g., classification of an animal on a picture). In the multilabel scenario, a single example can be assigned more than one class label (e.g., movie genres). We focus on the latter. For a particular set of inputs $\mathbf{x}$ (e.g., movie posters) and outputs $\mathbf{Y}$ (e.g., movie genre(s)), the risk formulation is the same as in (Menon et al., 2019):

$$R_{\mathrm{ML}}(\mathcal{F}) = \mathbb{E}_{(\mathbf{x}, \mathbf{Y})} \left[ \mathcal{L}_{\mathrm{ML}}(\mathbf{Y}, \mathcal{F}(\mathbf{x})) \right]. \tag{1}$$

The learning algorithm $\mathcal{F}$ is the estimand, the theoretical statistic. For one item $x_i$, the theoretical risk defines how close the estimand can get to that deterministic output vector $\mathbf{y}_i$. In practice, statistical models do output probabilities $\hat{\mathbf{y}}_i$ via an estimator and its estimate (also called propensies or suitabilities (Menon et al., 2019)). The solution to that stochastic-deterministic incompatibility is either to convert the estimator to a deterministic measure via decision thresholds (e.g., traditional cross-entropy loss), or to treat the estimand as a stochastic measure (our *sigmoidF1* loss proposal).

### 2.2 Estimator: the functional form

The estimator $f \in \mathcal{F}$ is any minimizer of the risk $R_{\mathrm{ML}}$. Predicting multiple labels per example comes with the assumption that labels are non mutually-exclusive.

**Definition.** *The multilabel estimator of $y_i^j$ is dependent on the input and other ground truth labels for that example, $\hat{y}_i^j = f(x, y_i^1, \ldots, y_i^{j-1}) = P(y_i^j = 1 | y_i^1, \ldots, y_i^{j-1}, x_i)$.*

By proposing this general formulation, we entrench that mutually-inclusive characteristic in the estimator. Contrary to Menon et al. (2019), our definition above models interdependence between labels and deals with thresholding for the estimate at training time for free. Waegeman et al. (2014) show that an estimator of an F-score can be used at inference time for multilabel classification, when using probabilistic models where parameter estimation is possible (e.g., decision trees, probabilistic classifier chains). When it is not possible, we resort to defining a loss function.

---

[2]https://github.com/gabriben/metrics-as-losses

### 2.3 Estimate: approximation via a loss function

Most of the literature on multilabel classification can be characterized as multilabel reductions (Menon et al., 2019): an approximation of the original multilabel problem via a loss function $\mathcal{L}(\mathbf{y}_i, f)$. It can take different forms.

**One-versus-all (OVA)** is a reformulation of the multilabel classification task to a sequence of $C$ binary classifications $(f^1, \ldots, f^C)$, with $C$ the number of classes, $\mathcal{L}_{\mathrm{OVA}}(\mathbf{y}_i, f) = \sum_{c=1}^{C} \mathcal{L}_{\mathrm{BC}}(y_i^c, f^c)$ where $\mathcal{L}_{\mathrm{BC}}$ is a binary classification loss (binary relevance (Brinker et al., 2006; Tsoumakas & Katakis, 2007; Dembczyński et al., 2010)), most often logistic loss. Minimizing binary cross-entropy is equivalent to maximizing for log-likelihood (Bishop, 2007, §4.3.4).

**Pick-all-labels (PAL)** gives the loss function $\mathcal{L}_{\mathrm{PAL}}(\mathbf{y}_i, f) = \sum_{c=1}^{C} y_i^c \cdot \mathcal{L}_{\mathrm{MC}}(y_i^c, f)$, with $\mathcal{L}_{\mathrm{MC}}$ a multiclass loss (e.g., softmax cross-entropy). In this formulation, each example $(x_i, \mathbf{y}_i)$ is converted to a multiclass framework, with one observation per positive label. The sum of inherently multiclass losses is used to represent the multilabel estimand.

Multilabel reduction methods are characterized by their way of reformulating the estimand, the resulting estimator, and the estimate. This allows the use of existing losses: logistic loss (for binary classification formulations), sigmoid or softmax cross-entropy loss (for multiclass formulations). These reductions imply a reformulation of the estimator (a.k.a. Bayes Optimal) as follows:

$$\hat{y}_i^j = f(x) = P(y_i^j = 1 | x_i). \tag{2}$$

Contrary to our definition of the original multilabel estimator (Section 2.2), marginal independence of label propensities is assumed. In other words, the loss function becomes any monotone transformation of the marginal label probabilities $P(y_i^j = 1 | x)$ (Dembczyński et al., 2010; Koyejo et al., 2015; Wu & Zhou, 2017). In literature reviews, the multilabel reductions OVA and PAL have been coined as *fit-data-to-algorithm*, as opposed to *fit-algorithm-to-data* (Zhang & Zhou, 2014), originally framed as *problem transformation* and *algorithm adaptation* respectively (Tsoumakas & Katakis, 2007)). For the purpose of our narrative, we propose the following formalization of this dichotomy: *fit-data-to-algorithm* formulates an additive loss over existing losses $\sum \mathcal{L}_C$, with $\mathcal{L}_C$ any classification loss and oftentimes a sum over all classes. This can be contrasted with *fit-algorithm-to-data*, where a custom loss $\mathcal{L}^*$ is built for the multilabel task. We further discuss this in Section 3 and Table 1.

### 2.4 Metrics: evaluation at inference time

There is consensus on the usefulness of a confusion matrix and ranking metrics to evaluate multilabel classification models at inference time (Koyejo et al., 2015; Behera et al., 2019; Wu & Zhou, 2017). Confusion matrix metrics come with caveats: most of these measures (i) require hard thresholding, which makes them non-differentiable for stochastic gradient descent; (ii) they are very sensitive to the number top labels to include $k$ (Chen et al., 2006); and (iii) they require aggregation choices to be made in terms of micro/macro/weighted metrics. Common confusion matrix metrics are Precision, Recall, F1-score or one-error-loss; see (Wu & Zhou, 2017) for others.

### 2.5 Multilabel estimate: F1 metric as a loss

A model's out-of-sample accuracy is commonly measured on metrics such as AUROC, F1 score, etc. These reflect an objective catered towards evaluating the model over an entire ranking. Due to the lack of differentiability, these metrics cannot be directly used as loss functions at training time (in-sample). Eban et al. (2017) propose a theoretical framework for deriving decomposable surrogates to some of these metrics. We propose our own decomposable surrogates tailored for multilabel classification (see Appendix A).

In a typical machine learning classification task, ground truth binary labels are compared to a probabilistic measure (or a reversible transformation of a probabilistic measure such as a sigmoid or a softmax function) (Bishop, 2007). If the number $n_i$ of labels to be predicted per example is known a priori, it is natural at training time to assign the $top_{n_i}$ predictions to that example (Lapin et al., 2016; 2015). If the number

Table 1: *SigmoidF1* and related loss formulations ordered by publication date. The solution column refers to our proposed formalization of the literature review on how to conduct multilabel classification: *D2A* refers to *fit-data-to-algorithm* (sum over existing or cross-entropy-like, CE-like, classification losses $\sum \mathcal{L}_\mathcal{C}$) and *A2D* refers to *fit-algorithm-to-data* (custom loss $\mathcal{L}^*$)

| Method | Solution | Model type | Context | Implementation | Surrogated metric | Modality |
|---|---|---|---|---|---|---|
| ACE [Fisher] | *D2A* | Any | Any | CE-like | – | Any |
| rankingLoss [Zhang & Zhou] | *D2A* | Any | Any | pair-rank | – | tabular |
| MFC [Huang et al.] | – | Gaussian mixtures | Mispronunciation detection | sigmoid | $F_1$ | Text |
| optLosses [Eban et al.] | *A2D* | Any | Any | – | $F_1$ | Theoretical |
| focalLoss [Lin et al.] | *D2A* | Neural net | Imbalanced-multiclass | CE-like | – | Image |
| deepF [Decubber et al.] | *A2D* | Neural net | multilabel | CE-like | $F_1$ | Image |
| softF1 [Chang et al.] | *A2D* | Neural net | Multilabel | Unbounded | $F_1$ | Image |
| ASL [Baruch et al.] | *D2A* | Neural net | Multilabel | CE-like | – | Image |
| RS@k [Patel et al.] | *A2D* | Neural net | Similarity | sigmoid | Recall | Image |
| polyLoss [Leng et al.] | *A2D* | Neural net | Imbalanced-multiclass, … | CE-like | – | Image |
| sigmoidF1 [ours] | *A2D* | Neural net | Multilabel | sigmoid | $F_1$ | Text & Image |

of labels per example is not known a priori, the question remains at both training and at inference time as to how to decide on the number of labels to assign to each example. This is generally done via a *decision threshold*, that can be set globally for all examples (Lipton et al., 2014). This threshold can optimize for specificity or sensitivity (Chen et al., 2006) – for per-class thresholding see Chu & Guo (2017). In Section 4, we propose an approach where this threshold is implicitly defined at training time, by using a loss function that penalizes explicitly for wrong label counts and fits to the original estimand in Definition 2.2: the F1 metric. In Section 4, we show how $F_1$ is formulated into a surrogate loss $\mathcal{L}_{\widetilde{F1}}$. Our contribution is thus in the continuation of the *fit-algorithm-to-data* trend, because we propose a custom loss function. That loss function is also the first to directly approximate the F1 score with non-divergent estimates (see Sections 4.1 and 4.2 on *boundedness*).

## 3 Related Work

In Section 2.3, we mentioned how existing solutions for multilabel tasks can be divided into *fit-data-to-algorithm* solutions, which map multilabel problems to a known problem formulation like multiclass classification, and *fit-algorithm-to-data* solutions, which adapt existing classification algorithms to the problem at hand (Madjarov et al., 2012). In most of this work, the term *multilabel classification* excludes *extreme* (tens of thousands of labels) (e.g., Jernite et al., 2017; Agrawal et al., 2013; Jain et al., 2019), *hierarchical* (parent and children labels) (e.g., Lehmann et al., 2015; Yang et al., 2019; Howard & Ruder, 2018) or *multiclass* (single label per example) subfields. These subfields call for their own solutions, including label embeddings (Bhatia et al., 2015) or negative mining (Reddi et al., 2019) for the *extreme* usecase.

**Fit-data-to-algorithm.** In fit-data-to-algorithm solutions, cross-entropy losses (Fisher, 1912; Good, 1952) are used at training time and thresholding is done at inference time to determine how many labels should be assigned to an instance. This has also been called multilabel reduction (Menon et al., 2019) and differs from multiclass-to-binary classifications (Zhang, 2004; Tewari & Bartlett, 2005; Ramaswamy et al., 2014). We can further distinguish between One-versus-all (OVA) and Pick-all-labels (PAL) solutions (Menon et al., 2019) (see Section 2). In OVA, one reduces the classification problem to independent binary classifications (Brinker et al., 2006; Tsoumakas & Katakis, 2007; Dembczyński et al., 2010; Wydmuch et al., 2018). In PAL, one reformulates the task to independent multiclass classifications (Boutell et al., 2004; Jernite et al., 2017; Joulin et al., 2017). The *label powerset* approach considers each set of labels as a class (Boutell et al., 2004). In Pick-One-Label (POL), a single multiclass example is created by randomly sampling a positive label (Joulin et al., 2017; Jernite et al., 2017). Alternatively, *ranking by pairwise comparison* is a solution where the dataset is duplicated for each possible label pair. Each duplicated dataset has therefore two classes and only contains instances that have at least one of the labels in the label pair. Different ranking methods exist (Zhang & Zhou, 2006; Hüllermeier et al., 2008; Loza Mencia & Furnkranz, 2008). Ranking loss has been shown to optimize for two Learning To Rank metrics (Chen et al., 2009). More recently, hierarchical datasets such as DBpedia (Lehmann et al., 2015) are used to fine-tune BERT-based models (Yang et al., 2019; Zaheer et al., 2020); the latter publications use cross-entropy to predict the labels.

**Fit-algorithm-to-data.** In fit-algorithm-to-data solutions, elements of the learning algorithm are changed (e.g., the back propagation procedure). Before focusing on the multilabel case, the multiclass literature has some examples of F1 surrogate loss functions in particular: in the context of SVMs, via pseudo linear functions (Narasimhan et al., 2015) or by learning a feasible confusion matrix (Narasimhan et al., 2015); in the context of deep networks, by learning the surrogate loss function via a dedicated neural network in the binary classification case (Grabocka et al., 2019), by optimizing performance measures composed of true positive and true negative rates (Sanyal et al., 2018) or via empirical utility maximization of F1 on 3-layer neural networks (Gai et al., 2019). Early representatives of multilabel fit-algorithm-to-data solutions stem from heterogenous domains of machine learning. MultiLabel $k$-Nearest Neighbors (Zhang & Zhou, 2007), MultiLabel Decision Tree (Clare & King, 2001), Ranking Support Vector Machine (SVM) (Elisseeff & Weston, 2001) and backpropagation for multiLabel learning with a ranking loss (Zhang & Zhou, 2006). More recently, the idea of multi-task learning for *label prediction* and *label count prediction* was introduced (ML$_{\mathrm{NET}}$, Du et al., 2019; Li et al., 2017; Wu et al., 2019). The literature has been clearly hinting at the usefulness of a single task loss function that approximates a metric. A formulation similar to our loss *unboundedF1* was proposed in an unpublished blog post, which was referred to as *softF1* (Chang et al., 2019). A similar proposal was to use the hinge loss as a decomposable surrogate for confusion matrix entries for binary classification (Eban et al., 2017). Outside of the context of neural networks, the *Maximum F1-score criterion* for automatic mispronunciation detection was proposed as an objective function to a Gaussian Mixture Model-hidden Markov model (GMM-HMM) (Huang et al., 2015). A recent paper used recall as a loss function for image similarity (Patel et al., 2022). In parallel, there is a growing consensus that the original cross-entropy loss (*fit-data-to-algorithm*) cannot solve all our problems. A variation of the cross-entropy loss adapted to multilabel classification has been proposed (Baruch et al., 2020; Wu et al., 2019); it extends the multiclass sparse class representation setting (Lin et al., 2017; Leng et al., 2022). In the ranking domain, LambdaLoss has been proposed to optimize directly for the lambdaRank metric (Wang et al., 2018). In the theoretical space, Eban et al. (2017) have proposed a generic framework for decomposable metrics, including $F1$ as a theoretical fractional linear program. Table 1 illustrates how *sigmoidF1* differs from the methods listed in this paragraph.

An important limitation shared by existing *fit-data-to-algorithm* and *fit-algorithm-to-data* approaches is the lack of a unified loss framework that deals with multilabel classification and can approximate a metric of interest. *sigmoidF1* computes an F1 surrogate loss over the aggregation of examples in a batch at training time.

## 4 Method

We introduce our approach for multilabel problems, with a smoothed confusion matrix metric as a loss (the original confusion matrix metrics rely on step functions and are therefore intractable, see for example the blue step function in Figure 2). We first briefly define our learning setting and define the confusion matrix metrics in this setting more formally.

We use the binary classification setting (two classes) to simplify notation, without loss of generalization to the multilabel case. In a typical binary classification problem with the label vector $\mathbf{y} = \{y_1, \ldots, y_n\}$, predictions are probabilistic and it is necessary to define a threshold $t$, at which a prediction is binarized. With $\mathbb{1}$ as an indicator function, $\mathbf{y}^+ = \sum \mathbb{1}_{\hat{\mathbf{y}} \geq t}$, $\mathbf{y}^- = \sum \mathbb{1}_{\hat{\mathbf{y}} < t}$ are thus the count of positive and negative predictions at threshold $t$. Let $tp$, $fp$, $fn$, $tn$ be number of true positives, false positives, false negatives and true negatives respectively:

$$
\begin{aligned}
tp &= \sum \mathbb{1}_{\hat{\mathbf{y}} \geq t} \odot \mathbf{y} \quad & fp &= \sum \mathbb{1}_{\hat{\mathbf{y}} \geq t} \odot (\mathbb{1} - \mathbf{y}) \\
fn &= \sum \mathbb{1}_{\hat{\mathbf{y}} < t} \odot \mathbf{y} \quad & tn &= \sum \mathbb{1}_{\hat{\mathbf{y}} < t} \odot (\mathbb{1} - \mathbf{y}),
\end{aligned}
\tag{3}
$$

with $\odot$ the component-wise multiplication sign. For simplicity, in the formulation above and the ones that follow scores are calculated for a single class, therefore the sum is implicitly over all examples $\sum_i$. This applies to the binary classification problem but also to our multilabel setting, when micro metrics are calculated (i.e., compute the metric value for each class, and then averaged over all classes). In the multilabel setting $\mathbf{y}$ can be substituted by $\mathbf{y}^j$ for each class $j$. Note that vectors could be trivially substituted by matrices

($\mathbf{Y}$) in Eq. 3 to obtain the macro formulation. Given the four confusion matrix quadrants, we can generate further metrics like precision and recall (see Table 5 in Appendix A). However, none of these metrics are decomposable due to the hard thresholding, which is, in effect, a step function (see Figure 2).

Next, we define desirable properties for decomposable thresholding, unbounded confusion matrix entries, and a sigmoid transformation that renders confusion matrix entries decomposable. Finally, we focus on a smooth F1 score.

### 4.1 Desirable properties of decomposable thresholding

We define desirable properties for a decomposable sign function $f(u)$ as a surrogate of the above indicator function $\mathbb{1}_{\hat{\mathbf{y}}<t}$.

**Property 1.** *Boundedness: $|f(u)| < M$, where $M$ is an upper and lower bound.*

The ground truth $\mathbf{y}$ is bounded between $[0,1]$ and thus it must be compared to a bounded prediction $\hat{\mathbf{y}}$, preferably bounded by $[0,1]$, to avoid further scaling.

**Property 2.** *Saturation: $\int_s^\infty f^{-1}(u) = \int_{-\infty}^{-s} f(u) = \epsilon$, with $\epsilon$ a number close to zero and $s$ a saturation bound.*

For the surrogate to be a proper sign function substitute, it is important to often return values close to 1 or 0. Saturation is defined in the context of neural network activation functions and refers to the propensity of iterative backpropagation to progressively lead to values very close to 0 or 1 after a long enough training period. Our aim is to reach that convergence quickly in order to force decisions towards 0 or 1 in order to be comparable to a step function. This highlights a tension: the sigmoid function should contrast outputs towards 0 or 1 but should not be too saturated, in order for the derivative at point $u$ to be non-null and information to flow back to the network (Krizhevsky et al., 2017).

**Property 3.** *Dynamic Gradient: $f'(u) \gg 0 \quad \forall\, u \in [-s, s]$, where $s$ is the saturation bound.*

Inside the saturation bounds $[-s, s]$, the derivative should be significantly higher than zero in order to facilitate stochastic gradient descent and backpropagation. Note that the upper and lower limits of $f(u)$ are interchangeably $[-1, 1]$ or $[0, 1]$ in this paper and in the literature. The conditions above still apply after linear transformation. Next, we show how our formalization of an unbounded F1 surrogate would not fulfill these properties and how our proposition of a smooth bounded alternative does.

### 4.2 Unbounded confusion matrix entries

A first trivial remedy to allow for derivation of the sign function $f(u)$, is to define *unbounded* confusion matrix entries by retaining the original logits (scores) when counting true positives, false negatives, etc. Countrary to the original confusion matrix definition in Eq. 3, $\overline{tp}$, $\overline{fp}$, $\overline{fn}$ and $\overline{tn}$ are not natural numbers anymore:

$$\begin{aligned}
\overline{tp} &= \sum \hat{\mathbf{y}} \odot \mathbf{y} & \overline{fp} &= \sum \hat{\mathbf{y}} \odot (\mathbb{1} - \mathbf{y}) \\
\overline{fn} &= \sum (\mathbb{1} - \hat{\mathbf{y}}) \odot \mathbf{y} & \overline{tn} &= \sum (\mathbb{1} - \hat{\mathbf{y}}) \odot (\mathbb{1} - \mathbf{y}),
\end{aligned} \tag{4}$$

where $tp$, $fp$, $fn$ and $tn$ are now replaced by rough surrogates. The disadvantages are that the desirable properties mentioned above are not fulfilled, namely (i) $\hat{\mathbf{y}}$ is unbounded and thus certain examples can have over-proportional effects on the loss; (ii) it is non-saturated; while non-saturation is desirable for activation functions (Krizhevsky et al., 2017), here it would be desirable to tend towards saturation (i.e., tend to values close to 0 or 1, so as to give the most accurate predictions at any thresholding values at inference time); and (iii) the gradient of that linear function is 1 and therefore backpropagation will not learn depending on different inputs at this stage of the loss function. However, this method has the advantage of resulting in a linear loss function that avoids the concept of thresholding altogether and is trivial to decompose for stochastic gradient descent.

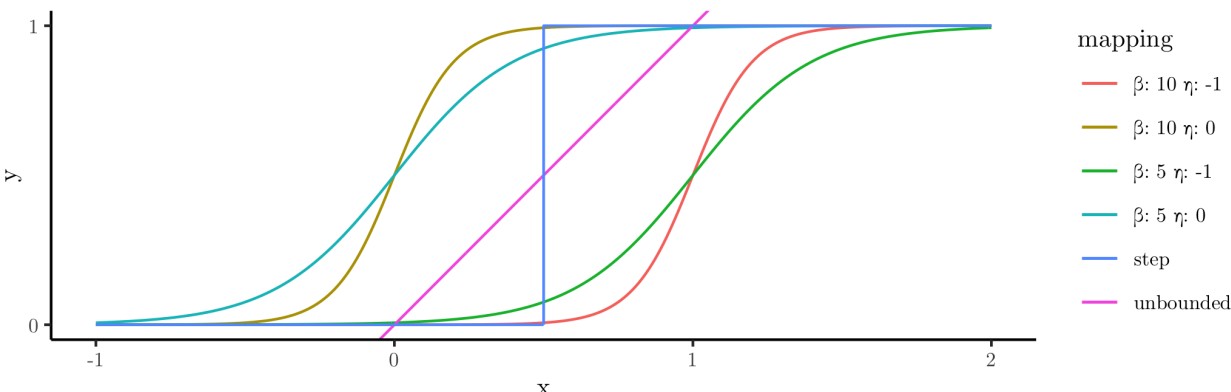

Figure 2: Different thresholding regimes: the step function (original *F1* metric) is not decomposable, the linear function is unbounded ($\mathcal{L}_{\overline{F1}}$) and tends to produce divergent gradients, whereas the sigmoid function ($\mathcal{L}_{\widetilde{F1}}$) is bounded and allows for differentiation due to its smooth curvature, tunable at different parametrizations.

### 4.3 Smooth confusion matrix entries

We propose a sigmoid-based transformation of the confusion matrix that renders its entries decomposable and fulfills the three desirable properties above:

$$\widetilde{tp} = \sum \mathbf{S}(\hat{\mathbf{y}}) \odot \mathbf{y} \qquad \widetilde{fp} = \sum \mathbf{S}(\hat{\mathbf{y}}) \odot (\mathbb{1} - \mathbf{y})$$
$$\widetilde{fn} = \sum (\mathbb{1} - \mathbf{S}(\hat{\mathbf{y}})) \odot \mathbf{y} \quad \widetilde{tn} = \sum (\mathbb{1} - \mathbf{S}(\hat{\mathbf{y}})) \odot (\mathbb{1} - \mathbf{y}),$$
(5)

with $\mathbf{S}(\cdot)$ the vectorial form of the sigmoid function $S(\cdot)$:

$$S(u; \beta, \eta) = \frac{1}{1 + \exp(-\beta(u + \eta))},$$
(6)

with $\beta$ and $\eta$ tunable parameters for slope and offset, respectively. Higher $\beta$ results in steeper slope at the center of the sigmoid and thus more stringent thresholding. At its extreme, $\lim_{\beta \to \infty} S(u; \beta, \eta)$ corresponds to the step function used in Eq. 3. Note that negative values of $\beta$ geometrically reflect the sigmoid function across the horizontal line at 0.5 and thus invert predictions. These smooth confusion matrix entries allow us to build any related metric (see Table 5 in Appendix A). Furthermore, the surrogate entries are decomposable, bounded, saturated and have a dynamic gradient.

### 4.4 Smooth macro F1 scores

F1 scores can be calculated on a macro and micro level. Macro-averaging regards all classes as equally important, whereas micro-averaging reflects within-class frequency. *unboundedF1* and *sigmoidF1* below are thought of as macro scores (aggregated over all classes). These scores require a high enough number of representatives in the four confusion matrix quadrants to learn from batch to batch. Ideally, each training epoch would have only one batch, so as to have the most representatives. Following Eq. 4, it is possible to define an *unbounded F1* score:

$$\mathcal{L}_{\overline{F1}} = 1 - \overline{F1}, \quad \text{where} \quad \overline{F1} = \frac{2\overline{tp}}{2\overline{tp} + \overline{fn} + \overline{fp}}.$$
(7)

While this alternative abstracts the thresholding away, which is convenient for fine-tuning purposes, it does not fulfill the desirable properties of a binarization threshold surrogate (see Section 4.2). *unboundedF1* will be used to benchmark against our proposed *sigmoidF1* loss. Given the definitions of smooth confusion matrix metrics above, we can now write $\mathcal{L}_{\widetilde{F1}}$:

$$\mathcal{L}_{\widetilde{F1}} = 1 - \widetilde{F1}, \quad \text{where} \quad \widetilde{F1} = \frac{2\widetilde{tp}}{2\widetilde{tp} + \widetilde{fn} + \widetilde{fp}}.$$
(8)

*sigmoidF1* is particularly suited for the multilabel setting because it is a proper hard thresholding surrogate as defined in the previous sections and because it contains a significant amount of information about label prediction accuracy: $\widetilde{tp}$, $\widetilde{fn}$ and $\widetilde{fp}$ are indicative of the number of predicted labels in each category of the confusion matrix but also contain a notion of certainty, given that they are rational numbers. The built in sigmoid function ensures that certainty increases along training epochs, as outlined by Property 2. Finally, as the harmonic mean of precision and recall (a property of F1 in general), it weighs in both relevance metrics.

In the next section, we implement Eq. 8 in PyTorch and TensorFlow as a custom loss as follows:

```
# with y the ground truth and z the outcome of the last layer
sig = 1 / (1 + exp(- β * (z + η)))
tp = sum(sig * y, dim=0)
fp = sum(sig * (1 - y), dim=0)
fn = sum((1 - sig) * y, dim=0)
sigmoid_f1 = 2*tp / (2*tp + fn + fp + 1e-16)
minimize(1 - sigmoid_f1)
```

The pseudocode above illustrates the elementwise multiplication of matrices $\mathbf{S}(\hat{\mathbf{y}})$ and $\hat{\mathbf{y}}$ over all examples in the batch and all possible classes.

## 5    Experimental Setup

We test multilabel learning using our proposed *sigmoidF1* loss function on four datasets across different modalities (image and text). For each modality we take a state-of-the-art model that generates an embedding layer and append a sigmoid activation and different losses. Multilabel deep learning is usually implemented with sigmoid binary cross-entropy directly on the last neural layer (a simplification of the OVA and PAL reductions). We follow this approach for our experiments (e.g., in (large) language models (Zaheer et al., 2020; Devlin et al., 2019)). Some baselines include multilabel reformulation choices: only keeping the top-$n$ occurring classes (often 4–10) (e.g., Zhang et al., 2015; Cunha et al., 2021), multiclass classification on each entity within an example (objects in an image, expressions in a text) (e.g., Lin et al., 2014; Wang et al., 2016; Wei et al., 2016; Zhu et al., 2017). We refrain from doing so.

Table 2: Descriptive statistics of our experimental datasets.

| Dataset | Type | Classes | Average label count | Number of examples |
|---------|------|---------|---------------------|--------------------|
| moviePosters | image | 28 | 2.2 | 37,632 |
| arXiv2020 | text | 155 | 1.9 | 26,558 |
| Pascal-VOC | image | 20 | 1.6 | 9,963 |
| MS-COCO | image | 80 | 2.9 | 122,218 |

### 5.1    Datasets

Table 2 lists the datasets we use. Two of the datasets are multilabel in nature. moviePosters is related to movies (Neha, 2018) and arXiv2020 relates to arXiv paper abstracts (Cornell-University, 2021). We use the image segmentation datasets Pascal-VOC (Everingham et al., 2007) and MS-COCO (Lin et al., 2014), with bounding boxes and one label per box. By attributing all box labels to the image as a whole, it has been used as a reference benchmark for multilabel classification. We refer to Appendix D for further descriptions of the datasets and references.

### 5.2    Learning framework

Our proposed learning framework consists of two parts: a pretrained deep neural network and a classification head (see Figure 1); different loss functions are computed in the classification head.

**Neural network architecture.**  For the moviePoster image dataset, we use a MobileNetV2 (Sandler et al., 2018) architecture that was pretrained on ImageNet (Deng et al., 2009). This network architecture is typically used for inference on small computing devices (e.g., smartphones). We use a version of MobileNetV2

already stripped off of its original classification head (Google, 2021). For the three text datasets, we use DistilBERT (Sanh et al., 2019) as implemented in Hugging Face. This is a particularly efficient instance of the BERT model (Huggingface, 2021). For the Pascal-VOC and MS-COCO datasets, we use the recent state-of-the-art resnet TresNet (Ridnik et al., 2021) pretrained on ImageNet (Deng et al., 2009) and some of the best practices for Pascal-VOC and MS-COCO collected in a recent benchmark (Baruch et al., 2020). We use TresNet-m-21K; 21K stands for Imagenet21K, the larger ImageNet corpus. In all cases, we use the final pre-trained layer as an embedding of the input. To ensure that the results of different loss functions are comparable, we fix the model weights of the pretrained MobileNetV2, DistilBERT and TresNet and keep the hyperparameter values that were used to be trained from scratch. At training time, we optimize with Adam for all three architectures and use In-Place Activated BatchNorm (Inplace-ABN) for TresNet (Rota Bulò et al., 2018).

The **classification head** is a latent representation layer (the final pretrained layer mentioned above) connected with a RELU activation. This layer is linked to a final classification layer with a linear activation. The dimension of the final layer is equal to the number of classes in the dataset. The attached loss function is either BCE (Binary Cross-Entropy), focalLoss (Lin et al., 2017), ASL (Baruch et al., 2020), unboundedF1 or sigmoidF1 (ours). When computing the loss at training time, a sigmoid transforms the unbounded last layer to a $[-1, 1]$ bounded vector that contrasts positive and negative predictions. These values are then used as inputs to any of the loss functions above over all classes and the entire batch of examples. In the case of $\mathcal{L}_{\widetilde{F1}}$, this corresponds to a surrogate macro F1. Given the vectorized computation of $\mathcal{L}_{\widetilde{F1}}$ (see Section 4.3), the computational burden is only marginally affected. At inference time, the last layer is used for prediction and is bounded with a sigmoid function. A threshold must then be chosen at evaluation time to compute different metrics. Figure 1 depicts this learning framework.

**Metrics.** In our experiments, we report on microF1, macroF1, Precision, mAP (used in some recent multilabel benchmarks; see Appendix A) and (micro-)weightedF1 (where within-class scores are weighted by their representation in the dataset). We focus our discussion around weightedF1 as it is the most comprehensive F1 measure we could find on multilabel problems: it is a micro measure, thus accounts for differences between classes, and has a reweighing argument, thus accounting for class imbalance. Given limited resources we rerun each model on each loss with 5 random seeds. With only 5 runs per loss function, hypothesis testing results would have been particularly sensitive to the choice of distribution.[3] Instead, we show the distribution of results in Appendix E, which show robust statistics (median and interquartile range). Note that cross-validation cannot be performed as Pascal-VOC and MS-COCO have fixed train-validation-test sets. There is an interaction between our optimization on *sigmoidF1* and our evaluation using (weighted) F1 metrics. We expect higher values on F1-related metrics during evaluation and thus report on alternative metrics too.

### 5.3 Hyperparameters and reproducibility

We implemented all losses in Pytorch and Tensorflow. Batch size is set at a relatively high value of 256 to increase accuracy over traditional losses (Smith et al., 2017), but also allow heterogeneity in the examples within the batch, thus collecting enough values in each quadrant of the confusion matrix (see Section 4.4 for a discussion). Regarding the *sigmoidF1* hyperparameters $\beta$ and $\eta$, we performed a grid search with the values in the range $[1, 30]$ for $\beta$ and $[0, 2]$ for $\eta$. In our experiments, we evaluate the sensitivity of our method to these hyperparameters (see Figure 2 and Appendix D for optimal values). We made sure to split the data in the same training, validation and test sets for each loss function. We trained for 60 (Pascal-VOC, MS-COCO) to 100 (arXiv2020, moviePosters) epochs, depending on convergence. Our code, dataset splits and other settings are shared to ensure reproducibility of our results.

## 6 Experimental Results

The goal of *sigmoidF1* ($\mathcal{L}_{\widetilde{F1}}$) is to optimize for the F1 score directly at training time in the context of multilabel classification. In this section, we test whether $\mathcal{L}_{\widetilde{F1}}$ can outperform existing loss functions on

---

[3] We found that, given some unstable results on unboundedF1, even a conservative student t distribution would imply that the 95% confidence interval covers metric values over 100%

multiple classification metrics. We present multilabel classification results for $\mathcal{L}_{\widetilde{F1}}$ on four datasets, moviePosters, arXiv2020, Pascal-VOC and MS-COCO in Table 3.

We recall Table 1, in which we highlight that $\mathcal{L}_{\text{BCE}}$ is originally designed for binary classification, $\mathcal{L}_{\text{FL}}$ for imbalanced multiclass, $\mathcal{L}_{\text{ASL}}$ to optimize mAP for multilabel classification. They are computed over each class at training time, as opposed to per batch for our $\mathcal{L}_{\widetilde{F1}}$ and $\mathcal{L}_{\overline{F1}}$. The latter two explicitly account for label dependencies in the loss function.

In general, Table 3 shows that $\mathcal{L}_{\widetilde{F1}}$ outperforms other loss functions on three possible formulations of the F1 metric (weightedF1, microF1 and macroF1). We also confirm that the recent ASL loss outperforms other losses on the precision and mAP metrics. $\mathcal{L}_{\widetilde{F1}}$ is designed as an F1 surrogate, it is thus not surprising for it to perform best on F1 metrics and comes at no noticeable additional computational cost (see Appendix C). We first analyze the F1 metrics before interpreting the precision and mAP results in more detail.

**Measured on the F1 metrics (weightedF1, microF1 and macroF1),** $\mathcal{L}_{\widetilde{F1}}$ and $\mathcal{L}_{\text{BCE}}$ always share the top 2 in performance, oftentimes far ahead of other losses. This highlights that losses inspired by BCE are not yet tailored to optimize for the F1 score in multilabel classification, and also that BCE is a good default choice in general. However, in certain settings, and in particular with our standard datasets Pascal-VOC and MS-COCO, $\mathcal{L}_{\widetilde{F1}}$ can provide clear improvements over the original BCE. macroF1 on the moviePosters dataset is a counter-intuitive exception to that observation: BCE outperforms $\mathcal{L}_{\widetilde{F1}}$ only on the macro measure, although $\mathcal{L}_{\widetilde{F1}}$ is essentially a macro F1 loss function, as it is calculated across all classes and over each entire batch. Similarly focalLoss is dominant on MS-COCO macroF1, but not significantly (see Figure 4). There is room for improvement on MS-COCO because we did not finetune the sigmoidF1 hyperparameters ($\beta$ and $\eta$) and instead reused the Pascal-VOC ones, due to resource constraints.

**On precision and mAP,** no top 2 losses emerge. Instead, results are dataset and modality dependent. Surprisingly, the traditional BCE loss outperforms other losses by far in precision on a thoroughly benchmarked dataset like Pascal-VOC. focalLoss delivers best results for MS-COCO on precision, probably because the original paper used MS-COCO as a benchmark to design their loss function (Lin et al., 2017). Precision performance gains are less clear on the two smaller datasets (arXiv2020 and moviePosters); $\mathcal{L}_{\overline{F1}}$ performs reasonably well.[4] Regarding mAP, $\mathcal{L}_{\text{ASL}}$ expectedly outperforms other methods on Pascal-VOC, confirming their own benchmarks (Baruch et al., 2020) and their ability to beat focalLoss and BCE on MS-COCO and PASCAL-VOC. Notably, $\mathcal{L}_{\text{ASL}}$ is also first on mAP on text data. This is the first time that ASL is tested on text data to the best of our knowledge. Overall, these mitigated results for precision and mAP motivate further research in optimizing directly for precision and mAP at training time.

**A note on thresholding and zero values.** For the bigger and more standard datasets Pascal-VOC and MS-COCO,[5] our neutral metric threshold of 0.5 provides results in line with the literature. With our own fine-tuning regime on a smaller model (see Section 5.2), our mAP scores are 1–2% away from the current state of the art (Baruch et al., 2020). On smaller datasets like arXiv2020, moviePosters and others (see Appendix F), the sigmoid activation per class at inference time are closer to zero. To a certain extent, this can be interpreted as the model having less confidence in its predictions (Guo et al., 2017). As a result, a neutral 0.5 threshold resulted in zero values on almost all losses and metrics for small datasets. Given the range of values in these predictions, 0.05 seems like the next best neutral threshold. We refrain from further finetuning the threshold for each dataset, loss and metric.[6] As a result of the absence of finetuning, moviePosters display zero values for $\mathcal{L}_{\text{FL}}$ and $\mathcal{L}_{\text{ASL}}$ on most metrics. This can be explained by the higher average label count for moviePosters. This is in opposition to the propensity of $\mathcal{L}_{\text{FL}}$ and $\mathcal{L}_{\text{ASL}}$ to deal with sparser label representation.

The analysis above highlights that sigmoidF1 can indeed optimize for F1 metrics (weightedF1, microF1 and macroF1) reliably and consistently, over six datasets in total (see Appendix E). Given the more mitigated

---

[4]$\mathcal{L}_{\overline{F1}}$ was found particularly unstable for Pascal-VOC over 5 different seeds (see the extended results in Appendix E). Provided it is unbounded, predictions can diverge towards (positive or negative) infinite values.

[5]The classes in Pascal-VOC and MS-COCO are a lot more concrete (e.g., car, person, bicycle) and are directly related to the original classes of ImageNet on which the TresNet and MobileNetV2 were trained, as opposed to movie genres for moviePosters or arXiv paper scientific domain.

[6]While optimizing the threshold at inference time is an interesting research topic, we refrain from doing so here, so as to disentangle the loss function benchmarking from the thresholding regime benchmarking.

Table 3: Multilabel classification mean performance in percent over 5 random seeds. The F1 metric variants are the focus here (weightedF1, microF1 and macroF1), since we aim to directly optimize for F1 at training time. precision and mAP are displayed for reference, as they are often used in the literature in that context. Metric are formally defined in Appendix A and thresholds are indicated there for each dataset. We reused fine-tuned Pascal-VOC sigmoidF1 hyperparameters ($\beta$ and $\eta$) for MS-COCO due to resource constraints.

| | **Loss** | **weightedF1** | **microF1** | **macroF1** | precision | mAP |
|---|---|---|---|---|---|---|
| TresNetm21K [2021] on MS-COCO @0.5 (CNN) | $\mathcal{L}_{\mathrm{BCE}}$[1912] | 79.02 | 75.81 | 79.55 | 82.52 | 81.21 |
| | $\mathcal{L}_{\mathrm{FL}}$[2017] | 81.28 | 79.18 | **81.76** | **85.73** | 84.88 |
| | $\mathcal{L}_{\mathrm{ASL}}$[2020] | 73.48 | 70.36 | 70.81 | 60.16 | **85.59** |
| | $\mathcal{L}_{\overline{F1}}$[ours] | 79.90 | 77.51 | 79.74 | 81.05 | 78.33 |
| | $\mathcal{L}_{\widetilde{F1}}$[ours] | **81.82** | **79.93** | 81.67 | 80.62 | 81.98 |
| TresNetm21K [2021] on Pascal-VOC @0.5 (CNN) | $\mathcal{L}_{\mathrm{BCE}}$[1912] | 87.52 | 85.85 | 87.76 | **90.75** | 91.54 |
| | $\mathcal{L}_{\mathrm{FL}}$[2017] | 72.54 | 59.24 | 76.82 | 84.70 | 76.19 |
| | $\mathcal{L}_{\mathrm{ASL}}$[2020] | 77.85 | 76.53 | 75.98 | 65.36 | **93.11** |
| | $\mathcal{L}_{\overline{F1}}$[ours] | 77.24 | 74.84 | 75.31 | 75.53 | 79.36 |
| | $\mathcal{L}_{\widetilde{F1}}$[ours] | **88.20** | **87.70** | **87.87** | 85.36 | 92.36 |
| DistilBert [2019] on arXiv2020 @0.05 (NLP) | $\mathcal{L}_{\mathrm{BCE}}$[1912] | 20.59 | 18.19 | 18.42 | 10.15 | 10.50 |
| | $\mathcal{L}_{\mathrm{FL}}$[2017] | 18.85 | 16.59 | 18.01 | 10.10 | 10.43 |
| | $\mathcal{L}_{\mathrm{ASL}}$ [2020] | 19.15 | 16.90 | 18.16 | **10.32** | **10.53** |
| | $\mathcal{L}_{\overline{F1}}$[ours] | 15.23 | 13.74 | 14.50 | 10.27 | 10.49 |
| | $\mathcal{L}_{\widetilde{F1}}$[ours] | **20.60** | **18.20** | **18.43** | 10.15 | 10.50 |
| MobileNetV2 [2018] on moviePosters @0.05 (CNN) | $\mathcal{L}_{\mathrm{BCE}}$[1912] | 13.79 | 9.47 | **12.94** | 5.51 | 5.78 |
| | $\mathcal{L}_{\mathrm{FL}}$[2017] | 0.00 | 0.00 | 0.00 | 0.00 | 5.80 |
| | $\mathcal{L}_{\mathrm{ASL}}$[2020] | 0.00 | 0.00 | 0.00 | 0.00 | 5.80 |
| | $\mathcal{L}_{\overline{F1}}$[ours] | 13.97 | 9.84 | 10.11 | **5.59** | **5.90** |
| | $\mathcal{L}_{\widetilde{F1}}$[ours] | **14.81** | **10.33** | 10.57 | 5.58 | 5.81 |

results for precision and mAP, it seems relevant to further explore opportunities of metrics-as-losses. Finally, BCE, which was designed with binary classification in mind, is a good first approximation.

**Sensitivity analysis.** In Figure 3, we show the sensitivity of *sigmoidF1* to different parametrizations of $\eta$ and $\beta$. Within the chosen values (see Section 5.3), we chose to display a parameter space similar to the one illustrated in Figure 2. Moving the sigmoid to the left allows the learning algorithm to tend to a (local) optimum. In general and across datasets, when sampling for $\eta$, we noticed how the optimum tended towards positive values. Offsetting the sigmoid curve to the left has the effect of pushing more candidate predictions to the rank of positive instance (or at least close to 1). We also note how $\beta$ (which cannot be negative or otherwise the sigmoid function would flip around the horizontal axis) is at best close to a value close to 0 on this dataset (we show discrete values here for display purposes). The sigmoid is thus relatively smooth, which involves dynamic gradients over different batches. The idea is similar to a high learning rate. In our experiments, this rarely gave rise to divergent behavior in the loss function (learning curve). We learn that it is necessary to tune hyperparameters for each dataset, as it is for $\mathcal{L}_{\mathrm{FL}}$, $\mathcal{L}_{\mathrm{ASL}}$ and others in Table 1.

The results in this section show that, in general, multilabel classification results measured on F1 metrics can be improved using sigmoidF1 – independently of the dataset, its modality or the neural network architecture.

## 7 Discussion

In multilabel classification, and more generally in the context of deep neural networks, losses are formulated to be decomposable for gradient descent. At inference time, however, end-users tend to look for clear-cut actionable decisions from the model (e.g., to automize the arXiv keywords selection, one needs to obtain a clear-cut set of keywords given each abstract). This is probably why most evaluation metrics in the

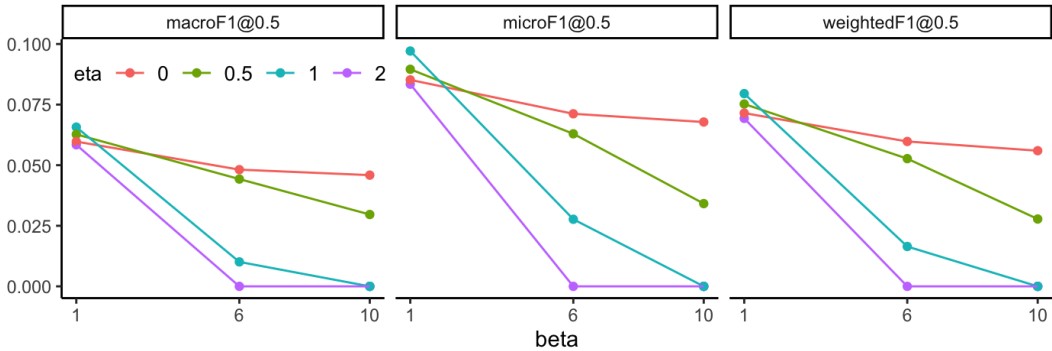

Figure 3: DistilBERT (NLP) on arXiv2020 – different weightedF1 scores at a 0.5 threshold for different values of $\eta$ and $\beta$ in a sampling region similar to Figure 2.

multilabel literature, with the notable exception of mAP, are also reliant on clear-cut counts (e.g. $tp$, $fn$, $fp$, $tn$). Although models are benchmarked on these values, we found little discussions on how to retrieve clear-cut counts from final softmax / sigmoid activations bounded by $[0, 1]$. Among our benchmarked losses, the authors of FocalLoss (Lin et al., 2017) use a global 0.5 threshold at inference-time. The authors of ASL (Wu et al., 2019) do not mention thresholding in the paper but a GitHub issue hints at the fact that they used 0.8 as a global threshold for MS-COCO.[7] We feel that defining clear-cut counts deserves more attention.

Such clear-cut counts are usually achieved via a *decision threshold*. Decubber et al. (2018) distinguish between *utility maximization* (at inference-time) and *decision-theoretic* (at training and at inference time) approaches.

**Utility maximization methods.** At inference time, a threshold can be set globally for all examples to optimize on the training data, before using it on the test data (Lipton et al., 2014; Decubber et al., 2018). This threshold can optimize for specificity, sensitivity (Chen et al., 2006) or directly for F1 (Decubber et al., 2018). Alternatively, different thresholds can be set per-class (Chu & Guo, 2017).

**Decision-theoretic methods.** Decision-theoretic methods operate both at training and at inference time. They stem from shallow learning fields and have multiple steps: (i) *encoding* the original item-label matrix to submatrices that each can be (ii) fit a traditional loss function (cross-entropy variations), before (iii) *decoding* the submatrices back to the original item-label matrix format via an inference-phase optimization solver. This methodology can be found across the shallow learning fields of SVMs (Ye et al., 2012), logistic regression (Dembczynski et al., 2011; Zhang et al., 2020), multinomial regression (Dembczynski et al., 2013), and Bayesian networks (Gasse & Aussem, 2016). These methods were implemented in a deep learning setting, where they had more success than *utility maximization* or fixed thresholding methods (Decubber et al., 2018). Decision-theoretic methods have at least 3 moving parts mentioned above and are thus complicated to benchmark against each other, let alone against inference-time thresholding or fix thresholding.

As hinted before with ASL and focalLoss, modern deep learning models tend to not tune the decision threshold, either with *utility maximization* or *decision-theoretic* methods. We propose to take a first step in this direction in the following.

For the sake of this discussion, we focus on the simplest *utility maximization* (inference-time) thresholding implementation. *Threshold Averaging* (Decubber et al., 2018) is a method that uses the training set to tune a global threshold, before applying it to the test set. Take $\hat{\mathbf{y}}_i$, a set of label predictions for one example $i$. We select each $\hat{y}_{ij}$ as a possible thresholding candidate to binarise the vector $\hat{\mathbf{y}}_i$. We then calculate instance-wise F1 scores over $\hat{\mathbf{y}}_i$. The value $\hat{y}_{ij}$ that results in the highest F1 score for an instance is chosen as the instance's threshold. This process is repeated for each instance $i$ in the training data. The average threshold over all instances in the training data is then chosen as the final global threshold for the test data.

---

[7]See https://github.com/Alibaba-MIIL/ASL/issues/8 and also insightful learning tricks at https://github.com/Alibaba-MIIL/ASL/issues/30

Table 4: Multilabel classification mean performance in percent over 5 random seeds. Global thresholds were found using a threshold-moving technique. Results are systematically lower than with a fixed threshold (see third row of Table 3). Metric are formally defined in Appendix A.

| | **Loss** | **weightedF1** | **microF1** | **macroF1** | precision | mAP |
|---|---|---|---|---|---|---|
| | $\mathcal{L}_{\mathrm{BCE}}$[1912] | 15.89 | 13.98 | 15.73 | 10.11 | 10.35 |
| DistilBert [2019] | $\mathcal{L}_{\mathrm{FL}}$[2017] | 16.40 | 14.14 | 17.22 | 9.83 | 10.42 |
| on arXiv2020 (NLP) – | $\mathcal{L}_{\mathrm{ASL}}$ [2020] | 17.49 | 14.86 | 17.77 | 10.33 | 10.51 |
| Threshold moving | $\mathcal{L}_{\overline{F1}}$[ours] | 16.52 | 14.27 | 16.70 | 9.98 | 10.43 |
| | $\mathcal{L}_{\widetilde{F1}}$[ours] | 15.11 | 13.19 | 15.20 | 10.05 | 10.41 |

In Table 4, we show results of *Threshold Averaging* (Decubber et al., 2018) on the arXiv dataset. It is notable here that ASL's mean results always outperform other losses. This time around, however, almost all boxplots IQRs intersect, thus results are very inconclusive (see Figure 8). We thus refrain from bold numbers like in Table 3. Most importantly, metrics are consistently below results from the original neutral fixed 0.05 threshold in Table 3. This is consistent with some of the results in (Decubber et al., 2018), showing that simple thresholding methods based on *utility maximization* are not sufficient to consistently beat fixed thresholds or *decision-theoretic* methods.

Inference-time decisions can completely change the outcome of a prediction set, of its resulting evaluation metrics, and, thus, even of the winning model. We hope that thresholding will be more broadly discussed in the future or at least for the thresholding method to be openly stated in research papers; we chose fixed neutral thresholds, to focus on the benchmarking of losses at training-time.

Together, utility maximization (inference-time) thresholding methods and decision-theoretic methods (at training and at inference time) form an under-explored research domain, with several open questions: (i) Which data split should be used for thresholding? With the entire training dataset (Decubber et al., 2018), there is a risk of overfitting the threshold. Maybe it is worth introducing a holdout set that is only used for threshold tuning. (ii) Should we threshold globally for interpretability or have a per-class or even per-instance threshold? (iii) Are *decision theoretic* (a.k.a. at training and at inference time) approaches also not prone to overfitting and are they efficient on large neural networks for large datasets? (iv) Can other losses than the classical cross-entropy loss be used to train *decision theoretic* models?

## 8   Conclusions

To solve multilabel learning tasks, existing optimization frameworks are typically based on variations of the cross-entropy loss. Instead – inspired by the binary classification literature (see most recently (Gai et al., 2019) and their F1 surrogate loss functions on 3-layer neural networks) – we propose the *sigmoidF1* loss, as part of a general loss framework for confusion matrix metrics. *sigmoidF1* loss can achieve significantly better results for most metrics on four diverse datasets and outperforms other losses on the weightedF1 metric. We thereby provide evidence that *sigmoidF1* is robust to modality, model architecture and dataset size, when optimizing for F1 metrics. Generally, our smooth formulation of confusion matrix metrics allows us to optimize directly for these metrics that are usually reserved for the evaluation phase. The proposed *unboundedF1* counterpart does not require hyperparameter tuning and delivered better results than existing multiclass losses on most metrics; it can act as a mathematically less robust approximation of *sigmoidF1*.

In future work and within the generic multilabel setting, a first incremental step could be to train on a bigger dataset like MS-COCO (Lin et al., 2014) (if provided with more resources) and use more robust transfer learning/finetuning procedures, for example with dynamic weight freezing for finetuning (Howard & Ruder, 2018). Alternatively, we could train a CNN or a BERT model for multilabel tasks with our smooth losses from scratch (cf., (Wu et al., 2019) and (Lin et al., 2017)). If training from scratch, this can be combined with representation learning (Milbich et al., 2020; Wang et al., 2020) or self-supervised learning, in order to model abstract relationships.

Next, we could validate if F1 or another confusion-matrix-metric-as-a-loss can tackle other multilabel settings, such as hierarchical multilabel classification (Benites & Sapozhnikova, 2015), active learning (Nakano et al., 2020), multi-instance learning (e.g., Soleimani & Miller, 2017; Zhou et al., 2012), holistic label learning (see dataset *Large Scale Holistic Video Understanding* (Diba et al., 2019)), or extreme multilabel prediction (Chang et al., 2020; Liu et al., 2017; Babbar & Schölkopf, 2017; Yen et al., 2017; Prabhu et al., 2018) (with missing labels (Yu et al., 2014; Jain et al., 2016)), where the number of classes ranges in the tens of thousands. Beyond the multilabel setting, *sigmoidF1* could be tested on any model that uses F1 score as an evaluation metric such as AC-SUM-GAN (Apostolidis et al., 2020).

One limitation of *sigmoidF1* is that it is computed at a macro level over the whole batch and ignores (micro) per class F1 scores. Given our limited GPU memory, we could not load enough examples in each batch to represent each confusion matrix quadrant of each class reliably. If such a route is followed, we could eventually finetune or learn $\beta_c$ and $\eta_c$ – the parameters of the sigmoid function – per class $c$.

We believe that smooth metric surrogates should inform future research on multilabel classification tasks. There is evidence of a growing interest in the literature (Chang et al., 2019; Huang et al., 2015; Patel et al., 2022) for metrics as losses and the objective of this paper is to further highlight their relevance, across modalities, architectures and dataset sizes. Based on the results presented in this paper, we consider metrics-as-losses (e.g., Jaccard, confusion matrix metrics, ranking metrics) as the next step in the evolution of multilabel classification algorithms.

### Acknowledgments

We thank our reviewers and the action editor for their valuable comments and suggestions, especially for pointing out to the thresholding literature, which led us to add a discussion section. We thank Nanne van Noord for his valuable suggestions on the paper structure and Maurits Bleeker for reviewing the text.

This research was (partially) funded by Bertelsmann SE & Co. KGaA; by the Hybrid Intelligence Center, a 10-year program funded by the Dutch Ministry of Education, Culture and Science through the Netherlands Organisation for Scientific Research, `https://hybrid-intelligence-centre.nl`.

All content represents the opinion of the authors, which is not necessarily shared or endorsed by their respective employers and/or sponsors.

### Author Contributions

All authors participated in the ideation process and discussions. Gabriel Bénédict was responsible for first drafts, the code and visualizations (except Figure 1, which is mostly attributable to Daan Odijk). Hendrik Vincent Koops, Daan Odijk and Maarten de Rijke revised initial drafts, proposed changes and feedback in content and form.

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

## Appendix A Evaluation Metrics

In our experimental evaluation, we consider a suite of metrics that are commonly used in the evaluation of multilabel classification to measure the effectiveness of multilabel prediction. These metrics are based on the confusion matrix and for which we provided smoothed surrogates to optimize directly (see Table 5).

Table 5: Confusion matrix with our proposed smoothed confusion matrix entries, $\widetilde{tp}$, $\widetilde{fp}$, $\widetilde{fn}$ and $\widetilde{tn}$ and six derived loss functions that use these smoothed confusion matrix entries. $\mathcal{L}_{\widetilde{F1}}$ is used in our experiments.

|  | Condition positive | Condition negative | $\mathcal{L}_{\widetilde{Accuracy}} = \frac{\widetilde{tp}+\widetilde{tn}}{\widetilde{tp}+\widetilde{fp}+\widetilde{tn}+\widetilde{fn}}$ |
|---|---|---|---|
| **Predicted positive** | True positive $\widetilde{tp} = \sum \mathbf{S}(\hat{\mathbf{y}}) \odot \mathbf{y}$ | False positive $\widetilde{fp} = \sum \mathbf{S}(\hat{\mathbf{y}}) \odot (\mathbb{1}-\mathbf{y})$ | $\mathcal{L}_{\widetilde{Precision}} = \frac{\widetilde{tp}}{\widetilde{tp}+\widetilde{fp}}$ |
| **Predicted negative** | False negative $\widetilde{fn} = \sum (\mathbb{1}-\mathbf{S}(\hat{\mathbf{y}})) \odot \mathbf{y}$ | True Negative $\widetilde{tn} = \sum (\mathbb{1}-\mathbf{S}(\hat{\mathbf{y}})) \odot (\mathbb{1}-\mathbf{y})$ | $\mathcal{L}_{\widetilde{NPV}} = \frac{\widetilde{tn}}{\widetilde{tn}+\widetilde{fn}}$ |
|  | $\mathcal{L}_{\widetilde{Recall}} = \frac{\widetilde{tp}}{\widetilde{tp}+\widetilde{fn}}$ | $\mathcal{L}_{\widetilde{Specificity}} = \frac{\widetilde{tn}}{\widetilde{fp}+\widetilde{tn}}$ | $\mathcal{L}_{\widetilde{F1}} = \frac{2\widetilde{tp}}{2\widetilde{tp}+\widetilde{fn}+\widetilde{fp}}$ |

When true positives and false positives are used, recall that $tp = \mathbb{1}_{\hat{\mathbf{y}} \geq t} \odot \mathbf{y}$ and $fp = \mathbb{1}_{\hat{\mathbf{y}} \geq t} \odot (\mathbb{1}-\mathbf{y})$, and thus a threshold $t$ must be set. For Pascal-VOC and MS-COCO, we set $t = 0.5$, as is commonly done in the early literature (Zhang & Zhou, 2014; Clare & King, 2001). In the recent literature, the chosen threshold at inference time can vary but was not found to be justified, we thus decide on neutral thresholds before training.

Extending $F_1$ to multiclass binary classification means deciding whether to pool classes. In a first pooled iteration, macro $F_1$ (Koyejo et al., 2015) equates to creating a single 2x2 confusion matrix for all classes:

$$F_1^{macro} = \frac{\sum^C 2tp_j}{2\sum^C tp_j + \sum^C fn_j + \sum^C fp_j}, \tag{9}$$

with $\sum^C(\cdot)$ as a short form of $\sum_{j=1}^{C}(\cdot)$, when summing over each class up to the $C$ classes. Micro $F_1$ (Lipton et al., 2014; Koyejo et al., 2015) amounts to creating one confusion matrix per class or unpooling:

$$F_1^{micro} = \frac{1}{C}\sum_{j=1}^{C} \frac{2tp_j}{2tp_j + fn_j + fp_j} = \frac{1}{C}\sum_{j=1}^{C} F_1^j. \tag{10}$$

Weighted micro $F_1$ (Behera et al., 2019) is similar but includes weighing to account for class imbalance, i.e., weighing each class by the number of ground truth positives:

$$F_1^{weighted} = \frac{1}{C}\sum_{j=1}^{C} p_j F_1^j \quad , \text{where } p_j = \sum_i \mathbb{1}_{\mathbf{y_i^j}=1}. \tag{11}$$

We also define micro precision

$$P^{micro} = \frac{1}{C}\sum_{j=1}^{C} \frac{tp_j}{tp_j + fp_j}. \tag{12}$$

mean Average Precision (mAP) has different definitions. We use mAP as defined for the MS-COCO and Pascal-VOC datasets (Padilla et al., 2020). Traditionally, Precision and Recall is computed over the intersection of object detection boxes. We use a slightly modified mAP (e.g., in (Baruch et al., 2020)), where precision and recall are computed over the predictions of labels on the whole image. We first obtain the average precision over each class:

$$\begin{aligned} \text{AP}_{\text{all}} &= \sum_i \left(R_{i+1} - R_i\right) P_{\text{interp}}\ \left(R_{i+1}\right) \\ P_{\text{interp}}\ \left(R_{i+1}\right) &= \max_{\tilde{R}:\tilde{R}\geq R_{i+1}} P(\tilde{R}), \end{aligned} \tag{13}$$

and then compute mean Average Precision:

$$\text{mAP}^{micro} = \frac{1}{C}\sum_{j=1}^{C} \text{AP}_i. \tag{14}$$

We write micro here to be explicit, but it seems to be mostly computed at the micro level in the literature.

## Appendix B    Focal Loss definition

We write down the *focalLoss* (Lin et al., 2017), as it deals specifically with class imbalance and is used as a baseline due to its popularity in the multiclass domain.

$$\mathcal{L}_{FL} = -\alpha^{\mathrm{j}} \left(1 - \hat{y}^{\mathrm{j}}\right)^{\gamma} \log\left(\hat{y}^{\mathrm{j}}\right), \tag{15}$$

with $\alpha^j$ and $\gamma$ hyperparameters. In the next section, we further specify the setup for focal loss and cross entropy as benchmarks for *unboundedF1* and *sigmoidF1*.

## Appendix C    Compute Time

Table 6 shows compute times in minutes for different on losses and different datasets on a single GPU *g4dn.12xlarge* AWS instance. The run-time is not particularly long, given that we freeze model weights of the pretrained image / text model.

Table 6: Average training time over 5 seeds in minutes (60 epochs for MS-COCO and Pascal-VOC, 100 epochs for the reminder two).

|  | MS-COCO | Pascal-VOC | arXiv2020 | moviePosters |
|---|---|---|---|---|
| $\mathcal{L}_{\text{BCE}}[1912]$ | 856 | 112 | 341 | 58 |
| $\mathcal{L}_{\text{FL}}[2017]$ | 851 | 108 | 428 | 59 |
| $\mathcal{L}_{\text{ASL}}[2020]$ | 856 | 109 | 427 | 59 |
| $\mathcal{L}_{\overline{F1}}[\text{ours}]$ | 858 | 116 | 381 | 58 |
| $\mathcal{L}_{\widetilde{F1}}[\text{ours}]$ | 858 | 111 | 351 | 52 |

## Appendix D    Experimental Setup Details

**moviePosters** consists of images of movie posters and their genres (e.g., *action*, *comedy*) (Chu & Guo, 2017).[8] The posters and labels have been extracted from IMDB and the dataset was previously used for per-class, post-training thresholding (see Section 3). The genre labels in this dataset are not mutually exclusive and of varying counts per movie.

**arXiv2020** is a subset of the newly created *arXiv dataset*[9] with over 1.7 million open source articles and their metadata. Our experiments use the abstracts and categories that are suitably non-mutually exclusive and of varying counts per example. The limited number of labeled classes render the older dataset unsuitable for our experiments. We write arXiv2020 for the subset of the *arXiv dataset* that only contains documents published in 2020. This results in around 26k documents. There is a longer history of using arXiv to create research datasets; the dataset we use is not to be confused with an earlier long document dataset that only features 11 classes (He et al., 2019), and was used in a recent long transformer publication (Zaheer et al., 2020).

**pascal-VOC and MS-COCO** stand for Pascal Visual Object Classes Challenge (VOC 2007) (Everingham et al., 2007) and Microsoft Common Objects in Context (Lin et al., 2014), respectively. They are object recognition/segmentation datasets. The earlier Pascal-VOC dataset has 20 possible object classes and around 10K examples. The later MS-COCO dataset has 80 possible object classes and around 200K class-annotated examples. Some multilabel classification literature for the image domain use object detection / segmentation datasets to perform multilabel classification:[10] MS-COCO, Pascal-VOC, NUS-WIDE, etc. (note that transformer models, which effectively distinguish the original objects on the image while predicting labels, perform better on this task (Liu et al., 2021)). Regarding Tresnet-m-21k (Ridnik et al., 2021), while an L and an XL version of the model exist, the code available online did not allow for correct loading of the weights.

We choose to ignore classes that are underrepresented, in order to give the model a fair chance at learning from at least a few examples. We define underrepresentation as a global irrelevance threshold $b$ for classes: any class $c$ that is represented less than $b$ times is considered irrelevant. We decided to set an irrelevance threshold $b$ on all datasets prior to conducting experiments, so as to not fine-tune for that feature. It was set to 1000 for both *arXiv2020* (145 of the original 155 classes remaining) and *moviePosters* (14 of the 28 classes remaining) and at 10 for *chemicalExposure* (all 38 classes remaining) and *cancerHallmarks* (all 33 classes remaining), in proportion to the number of classes and labels in each dataset. We used all classes for Pascal-VOC and MS-COCO since we are comparing with benchmarks that also do so.

**Hyperparameters.** For Pascal-VOC, we found $\{\beta = -0.75; \eta = 10.25\}$ to work best on weightedF1. Given the similarity of the two datasets and the potentially resource-hungry hyperparameter tuning of MS-COCO, we used the same hyperparameters for MS-COCO. For arXiv2020 and moviePosters, $\{\beta = 1; \eta = 9\}$ works best on weightedF1. These hyperparameters where tuned on the validation set and we report on the held out test set. It would be hard to give a general recommendation of hyperparameters, but it seems that $\{\beta = -0.75; \eta = 10.25\}$ is a good basis for image and that $\{\beta = 1; \eta = 9\}$ is a good basis for text.

**Setup.** We performed our experiments on Amazon Web Services cloud machines with data parallelization on up to 4 GPUs *g4dn.12xlarge*, with TensorFlow 2 (Abadi et al., 2015) and PyTorch (Paszke et al., 2019) as a gradient-descent backend.

---

[8]Labels at `https://tinyurl.com/y7ydyedu` and images at `https://tinyurl.com/y7lfpvlx`.
[9]Available at `https://tinyurl.com/5kypspya`
[10]See `https://paperswithcode.com/task/multi-label-classification`

## Appendix E   Extended Results

Table 3 shows our results as point estimates over 5 training random seeds. This section contains the distributional counterpart of Table 3, namely boxplots (Figure 4, 5, 6 and 7) with median and inter quartile range in the blue box. Figure 8 is the distributional counterpart of Table 4 (threshold-moving technique on the arXiv dataset) and outlines less conclusive results than for fixed thresholds.

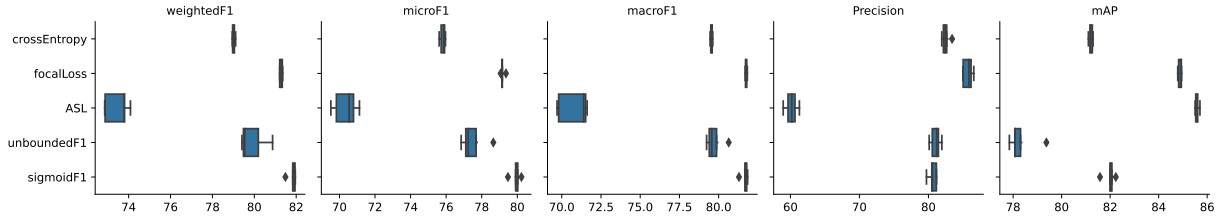

Figure 4: Tresnetm21K (CNN) on MS-COCO @0.5.

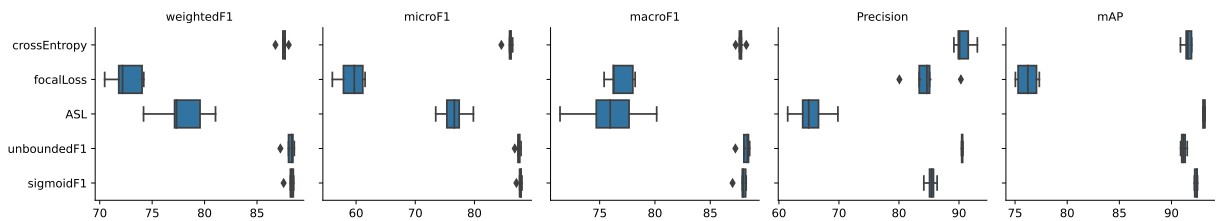

Figure 5: Tresnetm21K (CNN) on Pascal-VOC @0.5 (one outlier (<40) for unboundedF1 on each metric ignored for better visualization).

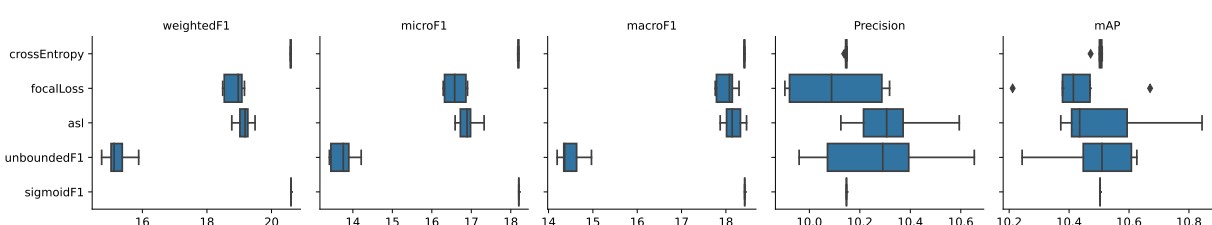

Figure 6: DistilBERT (NLP) on arXiv2020 @0.05.

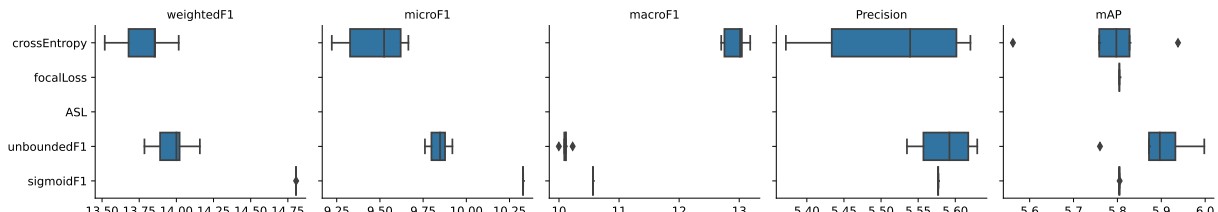

Figure 7: MobileNetV2 (CNN) on moviePosters @0.05 (zero values for focalLoss and ASL ignored for better visualization).

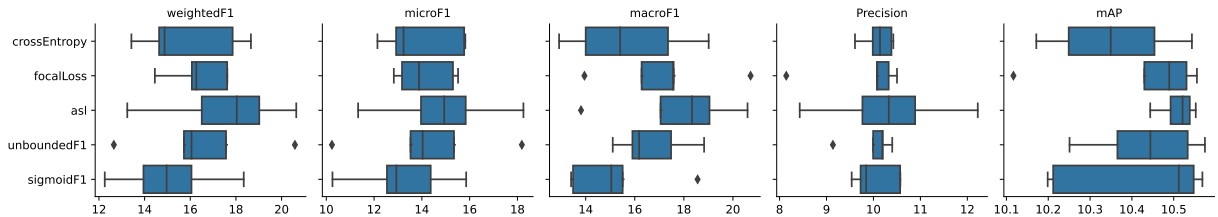

Figure 8: DistilBERT (NLP) on arXiv2020 – Threshold moving.

## Appendix F   Additional Experiments

This section details additional experiments on two further text datasets from the medical domain. Given that they are relatively small compared to our other benchmark datasets, we keep this discussion in the appendix. Table 8 illustrates the difference between our 4 main paper datasets and the 2 appendix datasets. Results on the latter are displayed in Tables 7a and 7b.

ML-NET (Du et al., 2019) has an interesting multitask approach to *fit-algorithm-to-data* methods for multilabel learning with unknown label count on text. The cancerHallmark (Hanahan & Weinberg, 2011)[11] and chemicalExposure (Larsson et al., 2014)[12] datasets were used. The third dataset diagnosisCodes could not be obtained (neither from the authors of ML-NET nor of the original paper (Perotte et al., 2014)). We aggregate sentence labels to the whole description for cancerHallmarks and chemicalExposure, as was done for ML-NET.

Table 7: Multilabel classification performance@0.05 on a single run.

(a) DistilBERT (NLP) + classification head on cancerHallmarks.

| Loss | weightedF1 | microF1 | macroF1 | Precision |
|------|-----------|---------|---------|-----------|
| $\mathcal{L}_{\text{BCE}}$ | 0.0 | 0.0 | 0.0 | 0.0 |
| $\mathcal{L}_{\text{FL}}$ | 10.8 | 19.0 | 4.4 | 7.1 |
| $\mathcal{L}_{\overline{F1}}$ | 17.0 | 17.6 | **9.8** | **8.9** |
| $\mathcal{L}_{\widetilde{F1}}$ | **20.2** | **31.3** | 9.5 | 5.9 |

(b) DistilBERT (NLP) + classification head on chemicalExposure.

| Loss | weightedF1 | microF1 | macroF1 | Precision |
|------|-----------|---------|---------|-----------|
| $\mathcal{L}_{\text{BCE}}$ | 5.1 | 5.8 | 1.2 | 4.7 |
| $\mathcal{L}_{\text{FL}}$ | 26.8 | 34.8 | 9.3 | 13.0 |
| $\mathcal{L}_{\overline{F1}}$ | 21.8 | 19.4 | **13.3** | **15.5** |
| $\mathcal{L}_{\widetilde{F1}}$ | **31.9** | **43.2** | 11.3 | 9.1 |

For arXiv2020, moviePosters, cancerHallmarks and chemicalExposure, we saw after a few preparatory training rounds that almost only *sigmoidF1* had non-zero results for $t = 0.5$. Class representation is a lot more sparse for these dataset, we thus set the evaluation metrics threshold to a reasonable value of 0.05 and train for 100 (arXiv2020, moviePosters) or 500 (cancerHallmarks and chemicalExposure) epochs until reaching convergence. Once thresholds were decided upon, no further threshold-hacking was performed. Note that a threshold of 0.8 on Pascal-VOC, as used by Baruch et al. (2020), does not alter the results.

On the smaller chemicalExposure and cancerHallmarks datasets (see Tables 7a and 7b respectively), the *unboundedF1* loss delivers good results for macroF1 and Precision and the *sigmoidF1* loss leads to higher

---

[11]Available at `https://github.com/sb895/Hallmarks-of-Cancer`

[12]Available at `https://github.com/sb895/chemical-exposure-information-corpus`

scores on the remainder of the metrics. We observe that *unboundedF1* scores higher than *sigmoidF1* on macroF1 on the two small text datasets (chemicalExposure and cancerHallmarks). Since *unboundedF1* forgoes thresholding altogether, we hypothesize that *unboundedF1* develops tolerance for sparse datasets with low number of class instances.

Table 8: Descriptive statistics of all datasets.

|                | Type  | Classes | Average label count | Number of examples |
|----------------|-------|---------|---------------------|--------------------|
| moviePosters   | image | 28      | 2.2                 | 37,632             |
| arXiv2020      | text  | 155     | 1.9                 | 26,558             |
| chemExposure   | text  | 38      | 6.1                 | 3,661              |
| cancerHallmarks| text  | 33      | 3.5                 | 1,582              |
| Pascal-VOC     | image | 20      | 1.6                 | 9,963              |
| MS-COCO        | image | 80      | 2.9                 | 122,218            |

Notably for the cancerHallmarks dataset, predictions from a model trained with cross-entropy do not reach high enough values to surpass the threshold and thus all metrics return zero values. This was further observed during experimentation, thus cross-entropy loss might not be a good fit for solving small-dataset multilabel problems.

