# OpenReview forum: "sigmoidF1: A Smooth F1 Score Surrogate Loss for Multilabel Classification"
_TMLR — Accepted by TMLR_

### Review · Reviewer_zFBc · 2022-06-18

**Summary Of Contributions:**

This paper proposes a new multi-label loss function called sigmoidF1 loss, which uses a sigmoid-based transformation of the confusion matrix that renders its entries decomposable and fulfills the three desirable properties of decomposable thresholding in multi-label learning.
.

**Broader Impact Concerns:**

None.

**Requested Changes:**

1. More datasets and comparison methods could be used for evaluation.

2. Minor issues. "classificaton" ==> "classification" "to a 100 epochs" ==> "to 100 epochs"


**Strengths And Weaknesses:**

Strengths:

1. The proposed sigmoid-based transformation fulfills three important and desirable properties of decomposable thresholding.

2. With the help of the proposed sigmoid-based function, training with the proposed loss function in multi-label learning is quite simple. As shown in the end of Section 4, only five lines of code are needed for calculation.

3. The experiments results show that the proposed sigmoidF1 loss can indeed optimize for F1 metrics.


Weakness:

It would be more convincing if more comparison methods could be used in experiments. There are many existing multi-label learning methods which could be considered as "fit-algorithm-to-data" methods.

---

> ### Author Response · Authors · 2022-06-22
> **on datasets and comparison methods**
>
> Thank you for your encouraging comments. Regarding the changes you request for more datasets and comparison methods for evaluation
>
> 1. Re: datasets. For the paper we selected datasets that we believe are representative of the text and image multilabel classification domains. We currently use 5 datasets (we report on 3 in the main body of the paper and on 2 in the appendix). We have recently discovered 3 more datasets that fit the multilabel setting (and the computing power that we have access to):
>    0. [Celebrity faces](https://mmlab.ie.cuhk.edu.hk/projects/CelebA.html "https://mmlab.ie.cuhk.edu.hk/projects/CelebA.html") for **image**, with labels like \[“glasses”, “beard”, …]. It is usually used for object detection (like PascalVOC), but could be used as a multilabel dataset.
>    1. Reuters for **text**; it is an old dataset but it could fit the bill.
>    2. PubMed for **text**; it is like the arXiv dataset that we use already but for medicine.
>
> If you believe that experiments on these datasets would add value, please let us know and we will include them. Also, if there are other datasets that you believe we should include, please let us know.
>
> 2. Regarding methods,  we found it hard to find “fit-algorithm-to-data” multilabel losses (i.e., not just a multiclass loss used for multilabel, like focal loss). We based our loss comparison on the [ASL loss paper](https://arxiv.org/pdf/2009.14119.pdf) from last summer and added two more losses, including ours. If you have more suggestions for “fit-algorithm-to-data” multilabel losses, please let us know.
>
> Finally, thank you for pointing out the typos; we will fix them.

---

> > ### Comment · Reviewer_zFBc · 2022-06-29
> > **Response to Response**
> >
> > Dear authors,
> >
> > Thanks for the response.
> >
> > For datasets, I suggest that you can add experiments on the datasets that you mentioned in the response. Furthermore, recently proposed multi-label classification methods are often evaluated on MSCOCO, which is a large scale image classification dataset. I believe that experiments on more datasets (especially on complicated datasets) could add value.
> >
> > For comparing methods, you can refer to the survey papers [1][2], in which the authors divide multi-label learning methods into two categories: **problem transformation** methods and **algorithm adaptation** methods. Specifically, problem transformation strategy could be considered as **fit-data-to-algorithm** and algorithm adaptation could be considered as **fit-algorithm-to-data**. You may find some representative algorithm-adaptation/fit-algorithm-to-data methods in the survey papers. Specifically, I think the ranking-based multi-label loss (e.g., method in [3]), which is also a natural choice for multi-label setting, could be used for comparison.
> >
> >
> > [1] Tsoumakas, Grigorios, and Ioannis Katakis. "Multi-label classification: An overview." International Journal of Data Warehousing and Mining (IJDWM) 3.3 (2007): 1-13.
> > [2] Zhang, Min-Ling, and Zhi-Hua Zhou. "A review on multi-label learning algorithms." IEEE transactions on knowledge and data engineering 26.8 (2013): 1819-1837.
> > [3] Zhang, Min-Ling, and Zhi-Hua Zhou. "Multilabel neural networks with applications to functional genomics and text categorization." IEEE transactions on Knowledge and Data Engineering 18.10 (2006): 1338-1351.
> >
> >
> > Reviewer zFBc

---

> > > ### Author Response · Authors · 2022-06-29
> > > **MSCOCO and losses for multilabel classification**
> > >
> > > # MSCOCO
> > > We figured that MSCOCO was the main missing dataset here. We have been running experiments in the past 5 days and have gotten the following results on one set of random seeds and the official MSCOCO train/test split ( metrics as columns):
> > >
> > > | loss         | weightedF1 | microF1   | macroF1   | Precision | mAP       |
> > > |--------------|------------|-----------|-----------|-----------|-----------|
> > > | unboundedF1  | 79.539383  | 77.242563 | 79.232018 | 80.111697 | 77.844354 |
> > > | sigmoidF1    | 81.956970  | 79.993927 | 81.846199 | 81.167993 | 81.984284 |
> > > | ASL          | 74.083145  | 71.121465 | 71.519348 | 61.301826 | 85.711580 |
> > > | focalLoss    | 23.751501  | 5.671050  | 34.300533 | 24.182463 | 30.362872 |
> > > | crossEntropy | 79.109749  | 75.863398 | 79.574142 | 81.932099 | 81.250341 |
> > >
> > > We were trying to avoid the costs of fine-tuning on MSCOCO 25 times (5 losses X 5 random seeds), but would regret the paper being rejected on these grounds. We will make sure to resubmit a version of the paper including MSCOCO within a few days.
> > >
> > > # losses for multilabel classification
> > >
> > > During the ideation and writing phase of this paper, [1,2,3] were helpful to recontextualise our methodology in historical methods. All three are discussed in our paper.
> > >
> > > We will come back with a more detailed answer for the ranking loss in [3]

---

> > > > ### Comment · Reviewer_zFBc · 2022-07-03
> > > > **Thanks for the response**
> > > >
> > > > Dear authors,
> > > >
> > > > Thanks for providing new experimental results. It seems that the proposed losses show more stable performance compared with ASL and Focal loss on MSCOCO, while standard CE loss also shows competitive performance on this dataset. Overall, the proposed losses show the superiority on multi-label tasks.
> > > >
> > > > We claim that although the empirical part is very important to evaluate the quality of a paper, we will not reject papers just due to the absence of a specific dataset.
> > > >
> > > > Best,
> > > >
> > > > Reviewer zFBc

---

> > > ### Author Response · Authors · 2022-07-05
> > > **ranking loss**
> > >
> > > It is helpful to be pointed out to other benchmarks that we omitted.
> > >
> > > In the Learning to Rank (LTR) literature, there is clear theoretical evidence that ranking loss can optimize for the metrics NDCG and MAP (https://proceedings.neurips.cc/paper/2009/file/2f55707d4193dc27118a0f19a1985716-Paper.pdf)[Chen et. al., 2009]. We could not find such evidence for F1. We also found out that the LTR community seems to be moving away from pairwise to listwise losses [Burges, 2010](https://www.microsoft.com/en-us/research/wp-content/uploads/2016/02/MSR-TR-2010-82.pdf).
> > >
> > > In the multilabel classification literature, we could not find a standard use of (pairwise) ranking loss or code for it.
> > >
> > > We do feel that using ranking loss for a multilabel classification problem is useful, but we think it is not in our scope of optimizing for F1 score in a multilabel setting.

---

### Review · Reviewer_rBR1 · 2022-06-20

**Summary Of Contributions:**

This paper studies multi-label learning and proposes *sigmoidF1* as a surrogate of the conventional F1 measure. The proposed approach is based on the confusion matrix, so it could be used for other measures defined on the confusion matrix such as precision and Jaccard index.

**Broader Impact Concerns:**

No concern.

**Requested Changes:**

Below are some detailed comments:

- To be rigorous, a loss is a function we want to *minimize*, so F1 cannot be a loss function. I would suggest to revise the wording throughout the paper.

- When talking about optimizing an objective function (e.g., F1) in a multilabel learning problem, we typically see two different setups: (a) optimization at the training time, and (b) optimization at the inference time. This paper belongs to the former. This point becomes clear, however, only late in the paper. I think it is much better if this is clearly stated early.

- Several variants of F1 measures are widely studied in the multilabel learning literature, for example, micro F1, macro F1, sample F1 (i.e., instance-wise F1). This paper addresses macro F1, but not the other variants. This should be specified, if not in the title, at least in the abstract or in the introduction.

- It isn't super clear what the difference between D2A and A2D (fit-data-to-algorithm vs. fit-algorithm-to-data) is. Is it correct to say that the key difference is if a multi-label objective (such as F1) is directly optimized during the training time? What is the significance to differentiate these two, if both share the same issue that you are trying to address? Two follow-up questions:
  - There is an existing categorization of multi-label learning approaches: problem transformation vs. algorithm adaptation (Tsoumakas, 2007). What does your categorization differ from the existing one?
  - In the description of fit-data-to-algorithm, it says that cross-entropy losses are used. This is inconsistent with Table 1, where many methods using other losses are labeled as D2A.

- "We focus our discussion around weightedF1 as it is the most comprehensive F1 measure we could find on multilabel problems." Could you explain why weightedF1 is the most comprehensive F1 measure? What do you mean by *comprehensive*?

- Since the proposed approach is based on the confusion matrix, can't it be also applied for other measures such as Jaccard or precision/recall with ease? I think it might be worth some discussions.

**Strengths And Weaknesses:**

Strengths:
 - The key idea of this paper is straightforward and easy to grasp.

Weaknesses:
- The setting of the proposed approach and its explanation need improvements.
- Overall, the writing needs polishing.
- As a journal paper, some aspects of the proposed approach aren't discussed in sufficient details.

---

> ### Author Response · Authors · 2022-06-22
> **miscellaneous corrections and the scoping of macroF1**
>
> Thank you for the detailed comments. One small clarifying comment:
>
> Our aim with this paper is to introduce the use of that sigmoid surrogate for different metrics. But since it is a new idea and our first paper on that idea, we felt like strictly scoping the topic F1 and the multilabel setting: we felt that the concept of using a non-decomposable metric as a loss requires some amount of justification already. This is important particularly for points 3 and 6 below.
>
> Regarding the requested changes,
>
> 1. We take your suggestion to mean that we should say that we maximize sigmoidF1 or minimize 1-sigmoidF1. This is indeed what we do in the code and we should have been clearer in the text. We will make this crystal clear the first time we introduce sigmoidF1.
> 2. Thank you for pointing it out. We will follow your suggestion and add “at training time” in the abstract and introduction.
> 3. That is a good point. Thanks. We will add the terms “surrogate loss for macro F1” in the abstract and introduction. We felt that sigmoidF1 was a handy descriptive term that generalizes over micro / macro / weighted. We have inspired colleagues to experiment on a micro version of sigmoidF1 for example, but they haven’t had conclusive results yet. Thank you for mentioning sample F1, we didn’t think of it and will explore if we can implement it as a metric.
> 4. D2A and A2D ([Zhang, 2013](https://ieeexplore.ieee.org/document/6471714 "A Review on Multi-Label Learning Algorithms")) are equivalents to problem transformation and algorithm adaptation,  respectively (Tsoumakas, 2007). The first time we introduce D2A and A2D we omitted to make that clear, unfortunately. We will do so, thank you for pointing it out. We chose D2A and A2D because the names felt more intuitive and were coined in a well cited paper. Regarding consistency and Table 1, our statement is that D2A often uses variations of the cross entropy loss (this is our observation, both Tsoumakas (2007) and Zhang (2013) have a more conceptual definition of D2A and A2D). This is the case for all D2A methods in Table 1 and is highlighted in the “implementation” column. We will make sure this is made clearer in the paper.
> 5. We agree, we will rephrase. weightedF1 is comprehensive because it is a micro measure, thus accounts for differences between classes, and has a reweighting argument, thus accounting for class imbalance.
> 6. We are confident that using sigmoid surrogates can be applied for any metric, including Jaccard, confusion metrics and even ranking metrics. We view our paper as a first step and focus on a single metric in order to have a clear message. We realize that we should strike a balance between discussing F1 only (justifying to only experiment on F1) and discussing all other metrics (justifying an enormous amount of experiments on all possible metrics). We focus on F1 but will expand the "broader implications" part of our conclusion.
>
> Finally, if you have specific suggestions to help us polish our writing (a weakness you mention), we would appreciate hearing from you .

---

### Review · Reviewer_2Ux7 · 2022-06-22

**Summary Of Contributions:**

This paper proposes a surrogate loss for the F1 metric or any other metric based on the confusion matrix for multi-label learning. Specially, it relaxes the step function with a threshold to a sigmoid-style loss with two tunable parameters. Such a loss function can be optimised directly by neural networks. It is a direct surrogate to the F1 metric, so it is estimated that a better F1 metric can be achieved in the testing phase. Experiments demonstrated that a better testing F1 is achieved on three datasets compared to other loss functions setting as the learning objective.

**Requested Changes:**

1. Add references [1-6]. Discuss and compare with them.

2. Add a theoretical study on how optimising the surrogate loss can impact the final performance on the non-differentiable loss (metric) F1, and study theoretically the impact of $\beta$ and $\eta$.

**Strengths And Weaknesses:**

##### Strong aspects

1. The paper focuses on an important question: finding a direct surrogate loss for the F1 metric which can be optimised directly by the neural networks.

2. The paper proposes a relaxation of the step function, in the same flavour as the sigmoid function, which is straightforward and easy to implement.

3. Some empirical results have demonstrated that the proposed surrogate loss could lead to better performance than other losses on F1.

#### Weak points

1. The biggest concern is the significance of the current study. There are many works focusing on optimising the F1 metric in general [1], or specially for multi-label learning [2,3,4]. Moreover, some recent works solve exactly the same research question as the current paper: proposing a surrogate loss function for the F1 metric which can be employed by neural networks in multi-label learning [5,6]. These works are highly related to the current study but have not been discussed or compared yet. Given that the same problem has been solved in [5], which provides an in-depth theoretical study, I do not see the significance in the current work, at least in the current version.
[1] Nan Ye, Kian Ming Adam Chai, Wee Sun Lee, Hai Leong Chieu. Optimizing F-measure: A Tale of Two Approaches. ICML 2012
[2] Krzysztof Dembczynski, Willem Waegeman, Weiwei Cheng, Eyke Hüllermeier. An Exact Algorithm for F-Measure Maximization. NIPS 2011: 1404-1412
[3] Krzysztof Dembczynski, Arkadiusz Jachnik, Wojciech Kotlowski, Willem Waegeman, Eyke Hüllermeier. Optimizing the F-Measure in Multi-Label Classification: Plug-in Rule Approach versus Structured Loss Minimization. ICML (3) 2013: 1130-1138
[4] Maxime Gasse, Alex Aussem. F-Measure Maximization in Multi-Label Classification with Conditionally Independent Label Subsets. ECML/PKDD (1) 2016: 619-631
[5] Mingyuan Zhang, Harish Guruprasad Ramaswamy, Shivani Agarwal. Convex Calibrated Surrogates for the Multi-Label F-Measure. ICML 2020: 11246-11255
[6] Stijn Decubber, Thomas Mortier, Krzysztof Dembczynski, Willem Waegeman. Deep F-Measure Maximization in Multi-label Classification: A Comparative Study. ECML/PKDD (1) 2018: 290-305

2. The proposed surrogate loss is a straightforward use of the sigmoid loss function in the step function in the F1 metric. Since this is a "surrogate" loss of the original (non-differentiable) loss function, it is highly recommended for the paper to do a theoretical study on how optimising the surrogate loss can impact the final performance of the desired loss. Specially, two hyperparameters are included in the proposed loss function to add more flexibility to the original sigmoid loss function. It is also interesting to see how such two hyperparameters can impact the final performance from a theoretical perspective.

---

> ### Author Response · Authors · 2022-06-22
> **missing references and theoretical study of the hypers eta and beta**
>
> Thank you for these valuable points. And thank you for the references that we will integrate in our table of literature (Table 1 in the paper).
>
> Regarding the requested changes more in detail,
>
> 1. We are in the process of reading these papers. We will clarify their relationship with our paper and include them in the literature table (Table 1 in our paper).
> 2. We had already been thinking about the sort of analysis that was done for ASL ([ASL](https://arxiv.org/pdf/2009.14119.pdf), section 2.5):  a gradient analysis on the hyperparameters. This consists in analytically deriving the gradient of the loss function and feeding a range of values through that function. ASL is an additive Cross-entropy like method (“As commonly done in multi-label classification, we reduce the problem to a series of binary classification tasks. » ([ASL](https://arxiv.org/pdf/2009.14119.pdf), section 2.1)), thus feeding values to the gradient function is more trivial because it can be done on a single hypothetical example. In the case of sigmoidF1, we compute the loss at a macro-level on the entire batch. Simulating values also implies adding assumptions on the distribution of the labels in the batch as a whole. We hope that such a gradient analysis would help towards a better theoretical understanding of the impact of beta and eta. We will attempt to make the gradient analysis in a fair manner, given the assumptions required on the distributions of all labels in a batch, as opposed to just looking at a single example like in Figure 3 in [ASL](https://arxiv.org/pdf/2009.14119.pdf.).

---

> ### Author Response · Authors · 2022-06-29
> **more on the literature [1,2,3,4,5,6]**
>
> We have thoroughly reviewed the literature that you suggested, we are thankful that you pointed them out. Some of us are from a statistical background and regret that [1,2,3,4] could not be found in the Deep Learning literature we consulted. Here is a short summary of how we understand this:
>
> # our understanding of the literature history
>
> The literature you mentioned is focussed on losses for SVMs [1], logistic [2] / multinomial [3] regression and bayesian networks [4]. As mentioned in [6], these are "MLC problems with shallow base learners that do not involve feature learning". This is also the case for [5], since it is a loss for frequentist linear logistic regression models (see, section 7.2 [5]). Thus we would not agree to say that [5] deals with neural networks. To summarise, [1,2,3,4,5] are shallow learners.
>
> [6] implements [1] and [3] in a modern deep learning architecture (VGG16) and benchmarks them against methods that involve a standard cross-entropy loss with threshold tuning. We assume that [2] and [4]  are implicitly covered by [6], since all [2,3,4] are Bayes optimal methods. [1,2,3,4] are therefore benchmarked (at least implicitly benchmarked for [2,4]) when implemented in DL in [6].
>
> Restricting the discussion to the best method found in [6], we focus on GFM_OR (the implementation of [3]). Looking at the code and the mathematical definition, we discovered that the "decision-theoretic" [6] method (I) reframes the problem with "plug-in classifiers" with a fit-data-to-algorithm method and (II) applies F-measure optimisation on the softmax outputs in the "inference phase" [6].
>
> This is an interesting approach, it implicitly models a joint distribution over the labels, thus fulfilling assumption (1) in our paper, as we do for sigmoidF1. We feel it deserves a study of its own where "plug-in classifiers" could be any of the existing losses used in the DL multilabel setting (i.e. BCE, FL, ASL, sigmoidF1, ...).
>
> This leaves [5] as the only unattended. [5] describes a formulation with $s$ the number of labels and $s^2+1$ multiclass logit losses. After the at-training-time loss minimisation, a second inference-time stage is performed. We note that we were not able to find code implementation for [5].
>
> Below, we go into the detail of our understanding of [5,6] in particular.
>
> ## our understanding of the literature methodology
>
> If we understand the work of [1,2,3,4,5,6] correctly, they all consist of a multi-phase approach.
>
> ### threshold-moving techniques:
>
> 1. classical binary / cross-entropy / logit loss
> 2. threshold the probabilistic outcome dynamically
>
> ### decision-theoretic techniques ([5] and the best performing approaches in [6]):
>
> 1. **encoding** "encode" the data from a 2D examples-labels matrix to a 3D matrix (e.g. into a one-hot-s-hot encoding, with $s$ the number of labels for GFM in [6] or using close to $s^2+1$ submatrices in [5]).
> 2. **model for 3D input-output** If a shallow (SVM/linear regression/bayesian networks) model is used, adapt it to that 3D input space (e.g. multinomial regression). If a deep model is used, adapt a classical pretrained model (like VGG16 [6]) to accept a 3D matrix as input and output. Below we restrain ourselves to the deep model context.
> 3. **loss** In the case of GFM (the best performing approach [6]) use an additive categorical cross-entropy loss on each of the $s$ 2D matrices. In the case of [5], use a logit loss.
> 4. **decode** at "inference phase" [6], "decode" that predicted 3D output matrix back into the original 2D format. This is an inference-time minimisation step that we understand to happen separately from the training-time loss minimisation step (both in the shallow [5] or deep learning context [6]).
>
> Focussing on decision-theoretic techniques (the best performing approaches in the DL study [6] and the most recent paper [5]), we think that there are 4 main moving parts that influence each other.
>
> We restrain our study to (3), as a study of loss functions to directly optimise for a metric with a single loss function and no change to data (1), model architecture (2) or the inference phase (4). We think only tweaking (3) is defendable for research purposes, as a way to focus on a domain, but also for production purposes: ASL seems to still be a state-of-the-art multilabel loss function.
>
> We also think that the decision-theoretic approach contains too many moving parts (the 4 above), to be compared to our sharply focused method. The loss function (3) could be BCE, ASL, sigmoidF1 etc. For the decoding phase (4), 4 methods seem to claim to be bayes optimal [2,3,4,5].
>
> To summarise, we restrain our paper to a focused predefined scope and improve the at-training-time F1 score in the lineage of deep learning multiclass classification techniques (BCE, Focal Loss, etc.) and its multilabel-specific alternative ASL.
>
> We thank the reviewers for pointing out the literature and will add it to our literature review.

---

### Author Response · Authors · 2022-06-22
**some context**

The motivation for our work on sigmoidF1 comes from an applied context of a video recommendation platform: our task was to classify movie posters into movie genres. When we started back in January 2021, we found very few resources available tailored to multilabel classification (i.e., a custom fit-algorithm-to-data loss). Since then, the [ASL](https://arxiv.org/pdf/2009.14119.pdf) loss was released last summer and it was the first practical loss function for multilabel classification we could find: a CE-like loss function (Focal loss with distinction between negative and positive samples) that performs well on the mAP metric (in our paper, we indeed recover the ASL SOTA for mAP on 2 / 3 datasets).

In parallel, with sigmoidF1 we propose a second fit-algorithm-to-data multilabel loss that optimizes for F1 score. In our paper we show that this works for micro / macro / weightedF1.

We realize that sigmoids and metrics have a great potential for generalization beyond macroF1 (microF1, sampleF1, Jaccard, confusion metrics, ranking metrics, etc.). In the paper, we report on the first step, a combination of macroF1 with a sigmoid function.

---

### Author Response · Authors · 2022-09-07
**Minor revisions version**

We thank the reviewers and editor for their insightful comments, that we addressed in the following way:

- We explicitly acknowledge the multiclass literature on F1 score surrogates in the intro and conclusion
- We added a discussion section to report on threshold-moving based methods from (Decubber et al. 2018) and compare to fixed thresholding results.
- We added (or when already present, discussed in more detail) the following refs:&#x20;
  - (Stijn Decubber, Thomas Mortier, Krzysztof Dembczynski, Willem Waegeman. Deep F-Measure Maximization in Multi-label Classification: A Comparative Study.)
  - (Dembczynski et al., 2010) (Wydmuch et al., 2018)
  - soft margin SVM reference
  - multiclasss non-decomposable performance measures: Narasimhan et al., "Optimizing Non-decomposable Performance Measures: A Tale of Two Classes"; Narasimhan et al., "Consistent Multiclass Algorithms for Complex Performance Measures"; Sanyal et al., "Optimizing non-decomposable measures with deep networks"
  - Grabocka et al., "Learning Surrogate Losses"
  - Gai et al., "Gradient-based learning for F-measure and other performance metrics"
  - (Eban et al., 2017)
- We added a discussion on good hyperparameters for image and for text separately in the appendix
- Some corrections we did:
  - Definition 1, remove double conditioning
  - Table 1, author names rather than years

---

### Decision · Action_Editors · 2022-08-02

**Recommendation:** Accept with minor revision

**Comment:**

The paper considers the problem of optimizing the multi-label F1 score with deep networks. The proposal is to construct a surrogate loss for this metric, by replacing the TP/FP rate terms with smooth (sigmoid-based) approximations.

A primary concern raised by reviewers was regarding the technical depth of the proposal. The idea of using a sigmoid approximation to the 0-1 loss has a long history in standard binary classification. The present paper essentially extends this idea to the multi-label setting, in a natural way (replacing each occurrence of TP/FP with a sigmoid surrogate). Given the large literature on principled approaches to F-score maximization, such critiques appear reasonable.

At the same time, the express purpose of TMLR is to judge papers based on whether the claims are well-supported, and whether a subset of the audience would find the results interesting. On these points, the paper is on better ground. To properly frame the contributions, the paper should more explicitly acknowledge upfront (e.g., in the Introduction and Conclusion) that the idea of using smooth approximations to the TP/FP rate being standard (see also references below), _but_ that the application of such an idea to multi-label problems has not been systematically explored, particularly in the case of modern deep networks. This would emphasize that the main message is that this strategy is practically effective for multi-label (not just binary) problems.

Further to the reviewers' detailed comments, below are a few additional comments from my own reading. The authors are suggested to incorporate these as appropriate.

**Threshold-based methods** The discussion period touched on threshold-based approaches as encountered in (Decubber et al., 2018), but apparently these were discounted owing to not being performant compared to the best-performing methods in that paper. This is fine if the present paper was then to compare against these best-performing methods, but it is aruged that "they include multiple moving parts and are therefore out of scope of a pure F1-score-as-a-loss paper". Given however that threshold-based methods are fairly simple to implement and tend to be competitive (if not the best-performing) baselines, it would be of interest to compare them against the present approach.

**Bibliography** It is appreciated that the authors incorporated reviewers' suggestions in the initial review phase. However, given the problem being considered is hardly new and features a vast supporting literature, it behooves the paper to properly situate itself with respect to relevant work; please see suggested references below.

- "Menon et al. (2019) define these as multilabel reduction techniques" -> I believe the idea (and terminology) of multilabel reductions is far earlier than this reference. e.g., (Dembczynski et al., 2010) provided one early analysis of the OvA reduction, and (Wydmuch et al., 2018) discuss multilabel reductions.

- "minimizing a step loss function is intractable (Reddi et al., 2019)" -> the latter paper seems an unusual choice for this citation; the idea of using surrogates to overcome the intractability of direct 0-1 loss minimisation has been explored since at least the soft margin SVM.

- missing many relevant citations on optimizing non-decomposable performance measures; e.g., Narasimhan et al., "Optimizing Non-decomposable Performance Measures: A Tale of Two Classes"; Narasimhan et al., "Consistent Multiclass Algorithms for Complex Performance Measures"; Sanyal et al., "Optimizing non-decomposable measures with deep networks" (amongst many others). These are mostly limited to the multi-class case, but there is strong relevance to the motivation of the present paper (which anyway relies on an appropriate reduction to binary or multi-class classification).

- a missing relevant citation is Grabocka et al., "Learning Surrogate Losses". This paper also discusses optimizing generic performance measures (for multi-class tasks), and takes the strategy of trying to *learn* a suitable surrogate loss for such a generic measure. The paper presents results with neural network architectures.

- another missing relevant citation is Gai et al., "Gradient-based learning for F-measure and other performance metrics". This paper also considers gradient-amenable surrogates for non-decomposable performance measures (in the multi-class setting).

**Hyperparameters** The method requires specifying two hyperparameters that control the shape of the sigmoid. These are chosen based on a holdout set. Different values appear to be optimal for different datasets. It would be ideal if a single choice was uniformly performant across datasets, since otherwise effectively one is using a different loss function for different tasks. Similar to Figure 3, it would be of interest to see whether there is a single choice that is uniformly performant (even if not optimal) across the datasets.

**Connection to (Eban et al., 2017)** The idea has quite some conceptual similarity to that of Section 3 in (Eban et al., 2017). The latter is for binary rather than multi-label problems, but it certainly appears as if the proposed algorithm is a direct translation of the idea of using a surrogate loss approximation to the TPR/FPR.

**Minor comments**

- Definition 1, the notation "P(yji | y1i, . . . , yj−1i = 1|x)" is confusing; why is there double conditioning?
- Table 1, it would be clearer to include author names rather than years; or, if space is a consideration, to just remove the year


### Summary of requested changes

- Add text in Introduction to acknowledge upfront that the idea of using smooth approximations to the TP/FP rate being standard (see also references below), *but* that the application of such an idea to multi-label problems has not been systematically explored, particularly in the case of modern deep networks

- Update bibliography with provided references, and discuss as appropriate

- Discuss similarity to Section 3 of (Eban et al., 2017)

- Discuss viability of using a consistent set of hyperparameters across datasets; potentially augment Figure 3 with such analysis

- Discuss more the viability of threshold-based methods; ideally, compare against these as well